# Mapping allosteric communications within individual proteins

Jian Wang [1], Abha Jain[2], Leanna R. McDonald[2], Craig Gambogi[2], Andrew L. Lee[2] & Nikolay V. Dokholyan [1,3,4 ✉]

Allostery in proteins influences various biological processes such as regulation of gene transcription and activities of enzymes and cell signaling. Computational approaches for analysis of allosteric coupling provide inexpensive opportunities to predict mutations and to design small-molecule agents to control protein function and cellular activity. We develop a computationally efficient network-based method, Ohm, to identify and characterize allosteric communication networks within proteins. Unlike previously developed simulation-based approaches, Ohm relies solely on the structure of the protein of interest. We use Ohm to map allosteric networks in a dataset composed of 20 proteins experimentally identified to be allosterically regulated. Further, the Ohm allostery prediction for the protein CheY correlates well with NMR CHESCA studies. Our webserver, Ohm.dokhlab.org, automatically determines allosteric network architecture and identifies critical coupled residues within this network.

[1] Department of Pharmacology, Penn State College of Medicine, Hershey, PA 17033−0850, USA. [2] Division of Chemical Biology and Medicinal Chemistry, Eshelman School of Pharmacy, University of North Carolina at Chapel Hill, Chapel Hill, NC 27599-7363, USA. [3] Department of Biochemistry and Biophysics, University of North Carolina at Chapel Hill, Chapel Hill, NC 27599, USA. [4] Departments of Biochemistry & Molecular Biology, Penn State College of Medicine, Hershey, PA 17033-0850, USA. ✉email: dokh@psu.edu

Since Nabuhiro Gō introduced his model of protein folding[1], whereby attractive amino acid interactions are assigned based on whether these residues are in proximity in the native state of the protein, it has been accepted that the native structure of a protein is determined to a significant extent by its folding pathway. If naturally selected sequences fold into unique structures, then both sequences and structures also possess information about folding dynamics, although this relationship remains an enigma. Directly related to this protein folding problem is the phenomenon of allostery[2], whereby perturbation at one site (the allosteric site) in a protein is coupled to a conformational change and/or dynamics elsewhere in the same protein. Perturbation at an allosteric site, induced by a stimulus such as phosphorylation[3], a point mutation[4], binding of a molecule[5], light absorption[6], or post-translational modification[7], can lead to changes in catalytic activity[8], structural disorder[9], or oligomerization[10]. Allostery regulates processes, including ligand transport[11] and metabolic function[12]. Allostery is an intrinsic property of all proteins: all protein surfaces[13] are potential allosteric sites subject to ligand binding or to mutations that may introduce structural perturbations elsewhere in the protein[14,15]. Drugs targeting allosteric sites could offer improved selectivity compared with traditional active-site targets[16,17], and allosteric pathways can be engineered to regulate protein functions[18–21]. We posited that dynamic couplings in atomic motions are related to protein structure and sought to predict allosteric coupling based solely on established protein structures with the goal of building maps of dynamic coupling in proteins without the use of expensive computational or experimental approaches.

Both experimental and computational methods have been proposed to identify putative allosteric sites and to study how perturbations at the allosteric site affect the active site[6,22–28]. Two different types of computational approaches have been developed[22,29,30]. One is based on molecular dynamics simulations or normal mode analysis, such as SPACER[31], which is based on elastic network model (ENM)[32]. The other one is based on information theory or spectral graph methods[22,33]. In recent years, there has been a surge in the number of studies of allostery using network models, a type of spectral graph methodology. For example, Amor et al.[22] proposed an atomistic graph-based calculation to reveal the anisotropy of the internal propagation of perturbations in proteins. Chennubhotla and Bahar[34,35] introduced a novel approach for elucidating the potential pathways of allosteric communication based on Markov propagation of information across the structure. Vishveshwara and coworkers[36] proposed protein structure graph (PSG) and then represented proteins as interaction energy weighted networks (PENs)[37] with realistic edge-weights obtained from standard force fields to identify stabilization regions in protein structures and elucidate the features of communication pathways in proteins. Atilgan et al.[38] proposed to calculate the average path lengths in weighted residue networks to analyze the perturbation propagation. They also introduced perturbation-response scanning (PRS)[39,40] to calculate the effectiveness and sensitivity of residues in propagating allosteric signals.

Based on physical considerations, here we develop a comprehensive platform for allosteric analysis, Ohm (http://Ohm.DokhLab.org). Ohm facilitates four aspects of allosteric analysis: (1) prediction of allosteric sites, (2) identification of allosteric pathways, (3) identification of critical residues in allosteric pathways, and (4) prediction of allosteric correlations between pairs of residues. For backward validation, we identify allosteric sites and pathways in Caspase-1 and CheY, and compare our predictions with known experimental results. We also validate Ohm on a dataset consisting of 20 proteins of known structure that are regulated allosterically. For forward validation, we determine all residue–residue correlations in CheY and compare these with nuclear magnetic resonance CHEmical Shift Covariance Analysis (NMR CHESCA) results. We also utilize mutagenesis to disrupt allosteric communication in CheY and compare changes in allosteric behavior with Ohm predictions. In sum, Ohm detects allosteric coupling in proteins based solely on their structures, enabling us to build maps of dynamic coupling in proteins without the need for expensive and time-consuming computational and experimental approaches.

## Results

**Implementation of a perturbation propagation algorithm.** A perturbation propagation algorithm is the foundation for allosteric network analysis in Ohm; this algorithm predicts allosteric sites, pathways, critical residues, and inter-residue correlations. The workflow of Ohm is illustrated in Fig. 1. The perturbation propagation algorithm is a repeated stochastic process of perturbation propagation on a network of interacting residues in a given protein. First, contacts are extracted from the tertiary structure of the protein. Next, the algorithm calculates the number of contacts between each pair of residues, and further divides the number of contacts by the number of atoms in each residue. This information, in turn, is used to obtain a probability matrix $P_{ij}$ (via Eq. (3) ("Methods")). Each probability matrix element, $P_{ij}$, is a measure of the potential that the perturbation from one residue is propagated to another residue. Next, the algorithm perturbs residues in the active site and the perturbation is propagated to other residues according to the probability matrix. At each step in the propagation, a random number between 0 and 1 is generated. If this random number is less than the perturbation propagation probability between residue $i$ and residue $j$, $P_{ij}$, then we propagate this perturbation from $i$ to $j$; if the number is greater than the perturbation propagation probability, the perturbation is not propagated. This process is repeated $10^4$ times. Finally, the frequency with which each residue is affected by a perturbation is calculated. We call this frequency the allosteric coupling intensity (ACI). Next, we cluster all residues according to their ACI values and their three-dimensional (3D) coordinates ("Methods"). Each cluster is predicted as an allosteric hotspot. To further utilize the perturbation propagation algorithm to identify allosteric pathways, in each step of the described perturbation propagation process, the residues through which the perturbation passes are recorded. This process yields the allosteric pathways that connect the active site with the allosteric site or sites.

**Validation of Ohm predictions.** The primary objective of the perturbation propagation algorithm is to identify candidate allosteric sites in proteins. We utilized a dataset compiled by Amor et al.[22], which consists of 20 known allosteric proteins (Supplementary Table 1), to test how accurately Ohm predicts the locations of allosteric sites. The dataset was compiled from SCOP database, which is a manually curated database using a hierarchical classification scheme to collect protein domains into structurally similar groups: $\alpha$, $\beta$, $\alpha/\beta$, $\alpha + \beta$, and multi-domain, which cover all the major fold-types for cytosolic proteins. They randomly selected proteins from each of the five major classes. The dataset includes seven monomers, two dimers, one trimer, seven tetramers, two hexamers, and one dodecamer proteins. The ligands of these proteins include chemical compounds, nucleosides, peptides, and DNA molecules. The lengths of proteins range from 147 to 3311 amino acids. Thus, the dataset covers a broad region of protein structure space. For each of the proteins, we used the 3D structure and the position of active sites as input and calculated the ACIs of all residues using Ohm. One of the

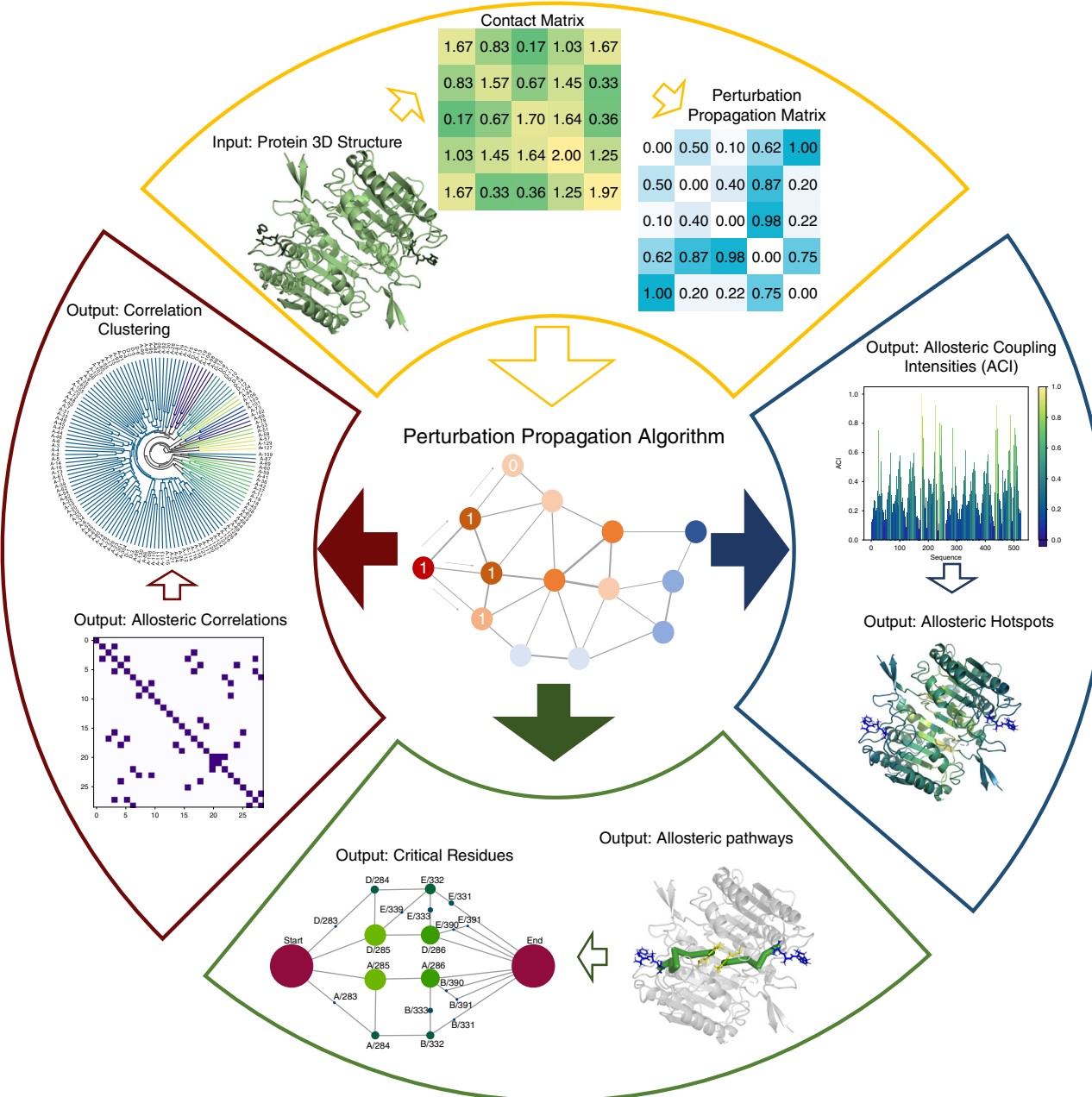

**Fig. 1 The Ohm workflow.** The Ohm workflow begins with input of the 3D structure of a protein. Next (moving clockwise), a contact matrix is calculated and a perturbation propagation probability matrix is generated according to Eqs. (1)–(3) ("Methods"). The perturbation propagation probability matrix provides the foundation of the perturbation propagation algorithm. If the position of the active site is known, the perturbation propagation algorithm calculates ACIs of all residues relative to the active site. We further devised a program to identify allosteric hotspots based on ACI values. If the position of both the active site and the allosteric site are known, the perturbation propagation algorithm can determine the allosteric pathways that connect the active site and the allosteric site. Further, Ohm identifies critical residues in allosteric pathways based on Eq. (4) ("Methods"). If neither the active site nor the allosteric site is known, the perturbation propagation algorithm can calculate allosteric correlations of each pair of residues. These correlations are then clustered to interrogate how residues in the protein are allosterically coupled.

proteins in the dataset is Caspase-1, a protein involved in cellular apoptosis and inflammation processes that is allosterically regulated[41]. Despite strong pharmacological interest, targeting the active sites of caspases with drug-like molecules has been very difficult. However, small molecules that bind to the allosteric sites of the protein have been shown to be potent inhibitors of caspase-1 enzymatic activity[42]. The structure of Caspase-1 contains two asymmetric dimers further organized into a tetrameric structure[43]. Each dimer contains one active site that is allosterically coupled to a distal active site[42]. With the tertiary structure of

Caspase-1 (PDB ID: 2HBQ) as input, Ohm was used to calculate atom contacts, construct the perturbation propagation probability matrix, and determine ACI values of all residues. There are 6 prominent peaks in ACI values along the sequence (Fig. 2a). The known allosteric site, which is at the center of the dimer, exactly corresponds to peak P6, indicating that the allosteric site of Caspase-1 was successfully identified by Ohm. In fact, these peaks are all around the allosteric site, and our clustering algorithm clustered them into one allosteric hotspot. We observed that residues having high ACI values typically have low solvent-

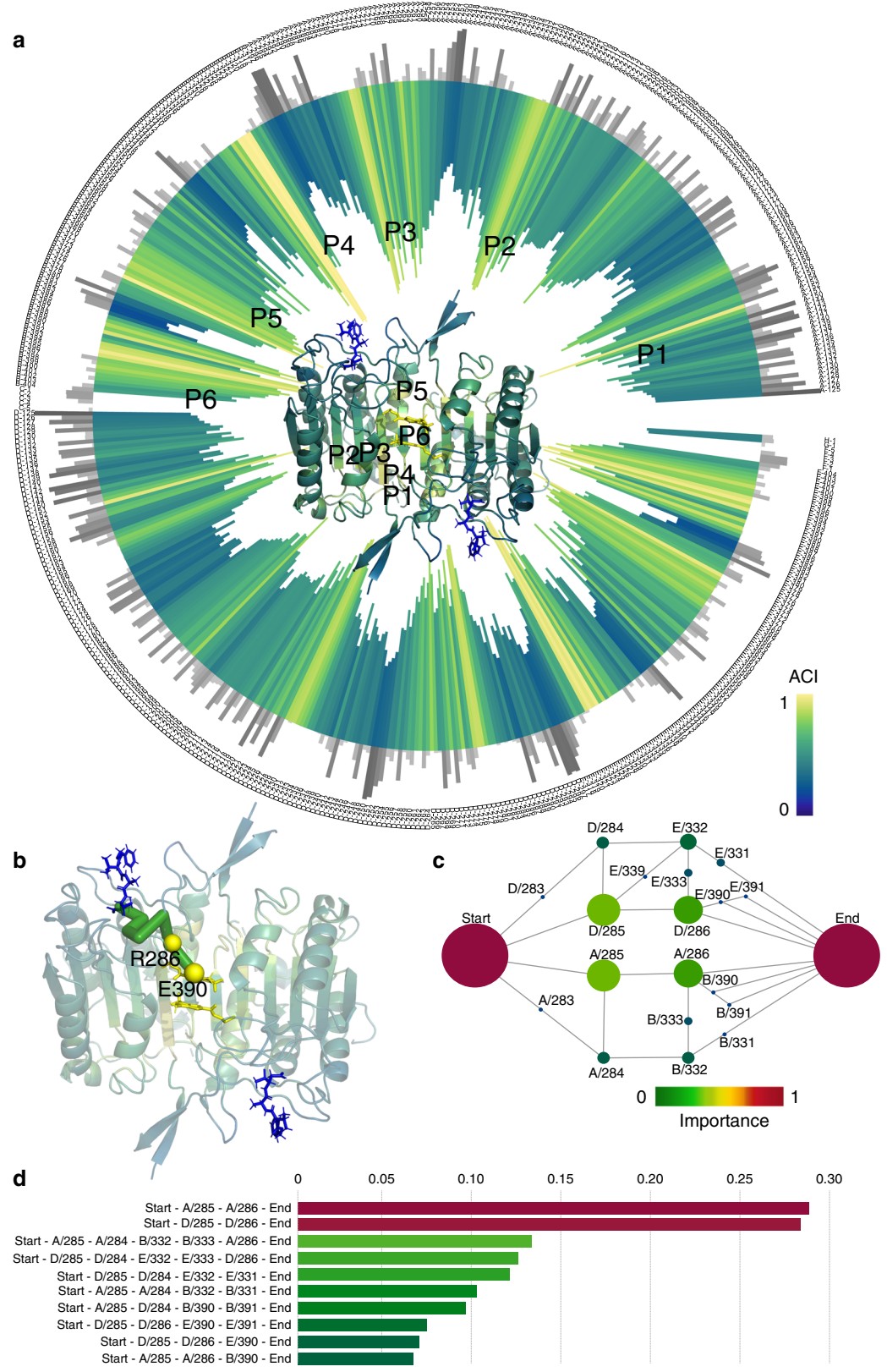

accessible surface areas (SASA) (Fig. 2a), and most low ACI areas are located on the surface.

To further evaluate how reliably Ohm predicts the allosteric pathways, we compared the predictions with previously reported experimental data. Datta and coworkers[41] mutated nine residues of Caspase-1 to test their roles in allosteric pathways

(Supplementary Table 2). Of these nine residues, R286A and E390A mutants strongly altered allosteric regulation, and S332A, S333A, and S339A had moderate effects, and S334A, S336A, S337A, and S388A mutants had little effect (Supplementary Table 2). Ohm predicted that R286 and E390 are the most critical residues in the Caspase-1 allosteric network (Supplementary

**Fig. 2 Allosteric analyses of Caspase-1. a** ACIs and SASAs of all residues in Caspase-1 (PDB ID: 2HBQ). The inward bar chart and the tertiary structure of CheY are colored by ACI of residues. Gray bars represent SASAs. In the tertiary structure of CheY, the active site ligand is colored by blue, and the allosteric site is at the center of the structure near P6. Peaks P1, P2, P3, P4, P5, and P6 of ACI are labeled both in the bar chart and the tertiary structure. **b** Allosteric pathway predicted by Ohm rendered as green cylinders in the 3D structure of Caspase-1. Yellow spheres are experimentally validated residues. **c** Critical residues in the allosteric pathways of Caspase-1 predicted by Ohm. The radius of each node indicates the importance of the residue in allosteric communication. Red color means high importance and green color means low importance. Each node is labeled by the chain name followed by a slash before the residue number. **d** Weights of 10 most important allosteric pathways of Caspase-1. The weights of the nodes in **c** and the pathways in **d** are illustrated in "Methods".

Table 2 and Fig. 2b, c); S332, S333, and S339 were the next most important residues, and S334, S336, S337, and S388 were the least important for allostery (Supplementary Table 2). These predictions are in perfect accord with experimental results. In addition, Ohm predicts that another residue C285, should be of importance to the network, although this has yet to be experimentally verified.

A second example of the accuracy of Ohm predictions is the 129-residue response regulator protein, CheY, which has allosteric properties and numerous crystal structures are available for both wild-type and mutant proteins[44–46]. The phosphorylation of D57 in CheY allosterically activates binding of FliM within bacterial flagellar motors[45,47,48]. There were four prominent peaks in the calculated ACI of residues in CheY (Fig. 3a). Remarkably, $BeF_3^-$ (the ligand binding in the allosteric site) and D57 (the allosteric site) have the highest and second highest ACI, respectively. The four ACI peaks P1, P2, P3, and P4 are all around the allosteric site, and they are finally clustered into one allosteric hotspot. Based on the analysis of pathways, the most important allosteric pathway is D57-T87-Y106-FliM (Fig. 3b–d). The two residues, T87 and Y106 (Fig. 3b), are of significantly higher importance than other residues based on ACI scores. Cho and coworkers[49] solved the NMR structure of activated CheY and proposed a Y–T coupling between residues Y106 and T87 to explain the mechanism of the allosteric activation of CheY: The phosphorylation of D57 causes T87 to move toward the phosphorylation site due to enhanced hydrogen bonding interaction between the two residues, which leaves more space for Y106 to occupy the buried rotameric state. The Y–T coupling is the reason for the importance of T87 and Y106 in the allosteric communication between the allosteric site and FliM, which is in agreement with our predictions. Thus, for both Caspase-1 and CheY, the residues predicted by Ohm to be the most critical components of the allosteric pathways in these two proteins are those previously established to be of high importance.

The perturbation propagation algorithm in allosteric pathways identification starts at the allosteric site, because the perturbation in protein is propagating from the allosteric site to the active site, but the perturbation propagation algorithm in allosteric site prediction actually starts at the active site, because the active site is known and the objective is to find the allosteric site. To interrogate the difference of perturbation propagation directions, we used the allosteric site D57 in CheY to predict the active site (Supplementary Fig. 1A). There are three major ACI peaks and the third one that includes residues 100-105 is exactly the active site. We have also identified the pathways from the active site to the allosteric site (Supplementary Fig. 1B). The most critical residues in the identified allosteric pathways are still 87 and 106. These results indicate that the allosteric correlation between the allosteric site and the active site in CheY is reversible, while the allosteric correlation in other proteins could also be irreversible[50].

We performed allosteric analysis for all 20 proteins (Fig. 4 and Supplementary Figs. 2–21) and compared the allosteric site prediction results to that of Amor's method (Supplementary Fig. 22 and Supplementary Table 4). We utilized the clustering algorithm ("Methods" section) to identify allosteric hotspots

based on ACI values and calculated the true-positive ratio (TPR) —the ratio of the number of true hotspots to the total number of predicted hotspots. Ohm identifies several allosteric hotspots for small proteins and less than 15 hotspots for large proteins (such as 1EYI, 6DHD, and 7GPB). In stark contrast, if we apply the clustering algorithm to the quantile scores, which is the metric in Amor's method to evaluate the allosteric correlation, the number of predicted hotspots is much larger than that predicted by Ohm (Supplementary Fig. 22a). A plethora of identified hotspots create hurdles for users to identify the true allosteric site. For large proteins such as 1D09, 1XTT, 1EFA, 7GPB, and 1YBA, >30 hotspots are identified based on quantile scores because the quantile scores are scattered around the structure (Supplementary Fig. 23). Most importantly, the TPR of hotspots predicted by Ohm is much higher than that predicted by Amor's method for most proteins in the dataset (Supplementary Fig. 22b). The average TPR of Ohm is 0.57, compared to 0.23 of Amor's method. TPR of Ohm-predicted hotspots for the four small proteins— 1F4V, 2HBQ, 1PTY, and 3K8Y—are all equal to 1. Besides, although 1XTT is a large tetramer protein composed of 868 residues, the TPR of Ohm is still equal to 1. We also calculated the positive predictive value (PPV)—the ratio of the number of identified allosteric site residues to the total number of all allosteric site residues—of Ohm and Amor's method, respectively (Supplementary Fig. 22c). Ohm can recapitulate more allosteric site residues than Amor's method for most proteins. The PPV of Ohm is 0.72, compared to 0.48 of Amor's method. These results indicate that Ohm outperforms Amor's method in the ability to identify allosteric sites by featuring both higher TPR and higher PPV.

The pathways are identified by the perturbation propagation algorithm and the critical residues are identified by the importance in the pathways according to Eq. (4). Dijkstra's algorithm[51] has also been utilized by network models to identify optimal pathways[38] and betweenness centrality is usually utilized to measure the importance of nodes in the network. For each of the 20 proteins, we compared the top-10 ranked residues identified by importance of residue in Ohm pathways to residues identified by betweenness centrality in Ohm pathways and residues identified by betweenness centrality in Dijkstra optimal pathways (Supplementary Table 5). The three methods share some common residues but these residues are ranked differently. For CheY (1F4V), the most critical residues identified by Ohm are 87 and 106, while these two residues are lower-ranked by the other two methods.

Finally, we utilized a designed four-helix bundle protein (1MFT) as the negative control (Supplementary Fig. 24). The tertiary structure colored by the calculated ACI values shows that apart from the designated pseudo active site—the N-terminus— there are no ACI hotspots in this un-allosteric protein structure.

**Comparison of allosteric correlations from CHESCA and Ohm.** Taking advantage of the celerity of ACI calculation, Ohm could yield the ACIs of all residue–residue pairs of even a large protein in a short period of time. Experimentally, inter-residue

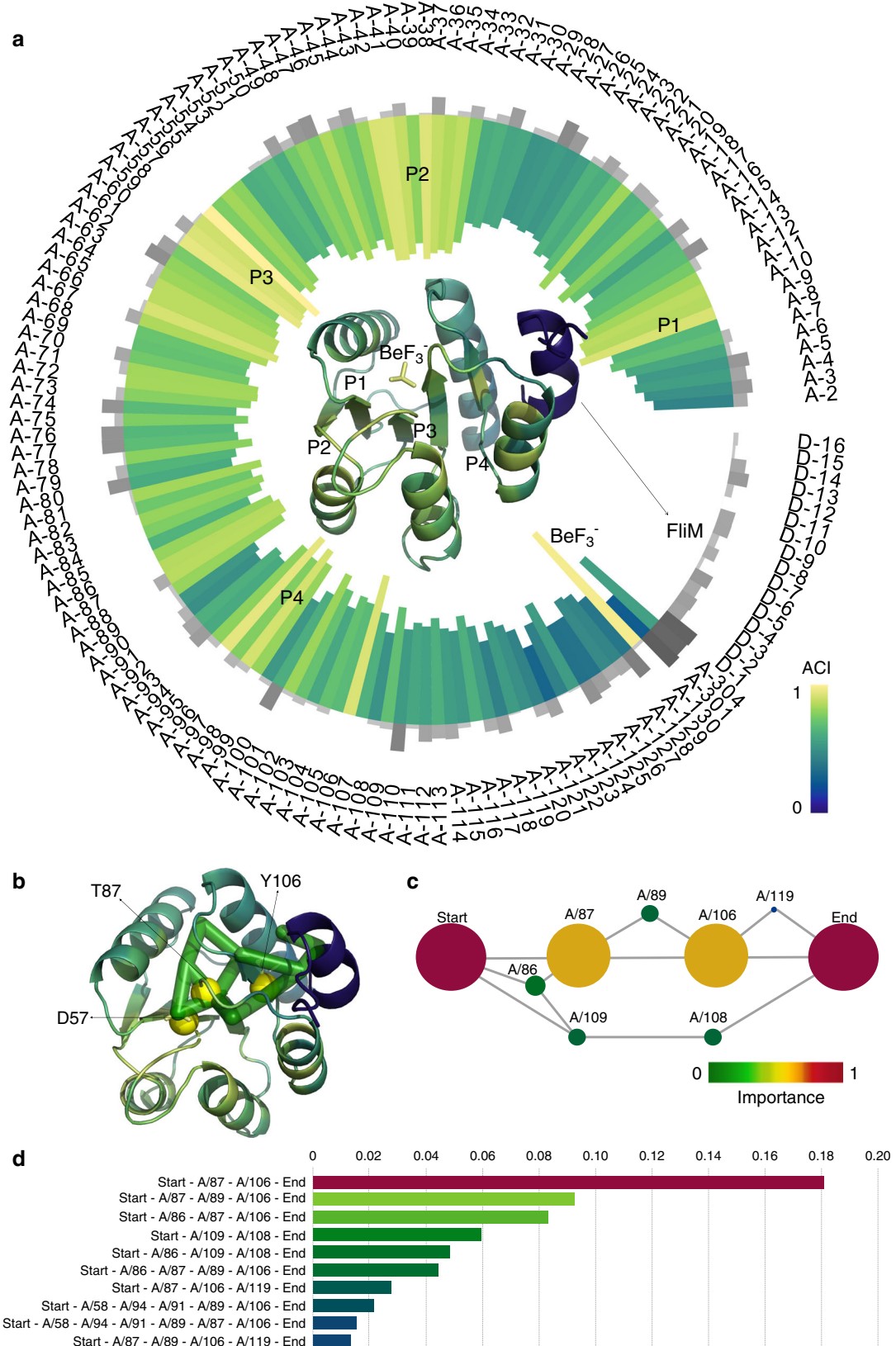

allosteric correlations can be determined using CHESCA based on the combination of agglomerative clustering and singular value decomposition as proposed by Selvaratnam et al.[52]. Aimed at revealing extended networks of coupled residues, CHESCA is essentially a covariance analysis of NMR chemical shift changes caused by local perturbations, such as a series of related compounds or mutations[53,54]. We analyzed seven different mutants of unphosphorylated CheY (F8V, D13K, M17A, V21A, T87I,

**Fig. 3 Allosteric analyses of CheY. a** ACIs and SASAs of all residues in CheY (PDB ID: 1F4V). The inward bar chart and the tertiary structure of CheY are colored by ACI of residues. Gray bars represent SASAs. In the tertiary structure of CheY, the active site ligand (FliM) is colorerd by blue, and $BeF_3^-$ is the ligand in the allosteric site. The four peaks P1, P2, P3, and P4 of ACI are labeled both in the bar chart and the tertiary structure. **b** Allosteric pathway predicted by Ohm rendered as green cylinders in the 3D structure of CheY. Yellow spheres are experimentally validated residues. **c** Critical residues in the allosteric pathways of CheY predicted by Ohm. The radius of each node indicates the importance of the residue in allosteric communication. Red color means high importance and green color means low importance. Each node is labeled by the chain name followed by a slash before the residue number. **d** Weights of ten most important allosteric pathways of CheY. The weights of the nodes in **c** and the pathways in **d** are illustrated in "Methods".

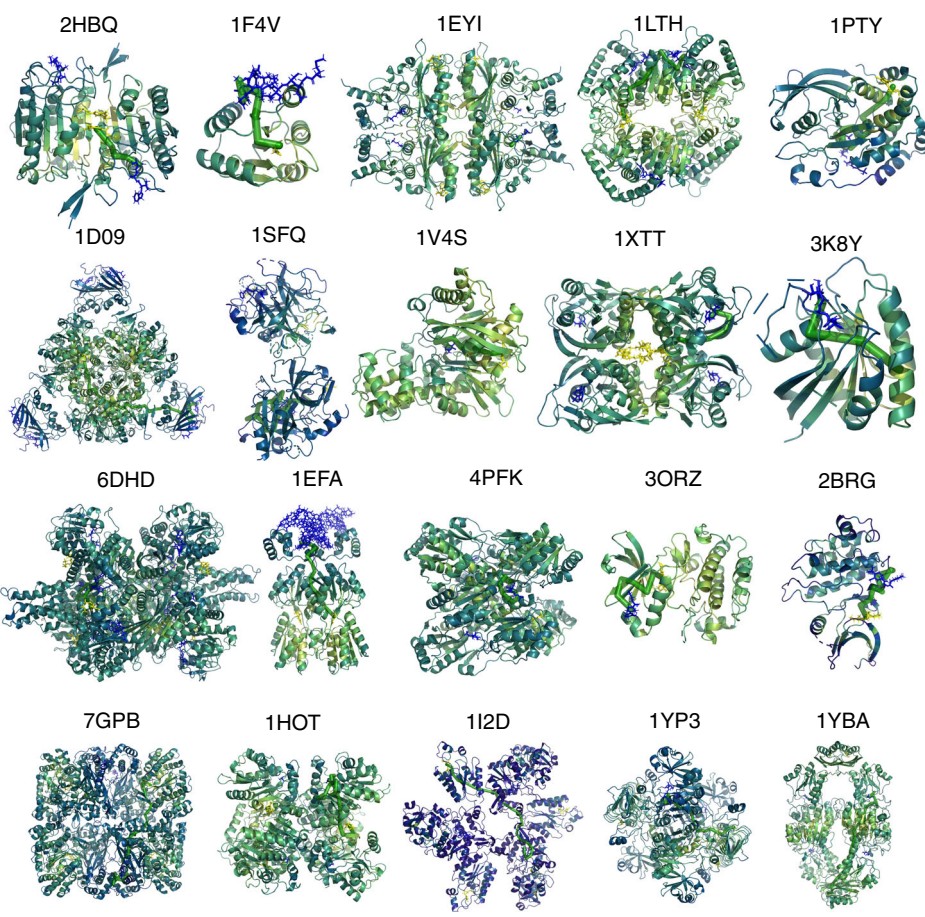

**Fig. 4 Tertiary structures of all 20 proteins colored by ACI.** Ligands in active sites are colored by blue, and ligands in allosteric sites are colored by yellow. One of the highest weight pathways identified by Ohm is shown as green consecutive cylinders for each protein. Resides with high, moderate, low, and extremely low ACI values are colored by yellow, green, and blue, respectively. Ohm analysis results for all 20 proteins are provided in Supplementary Data 3.

Y106W, and A113P) using CHESCA (see "Methods"). Inter-residue cross-correlations are determined for residue pairs (Fig. 5a). As there are errors in the calculated allosteric correlations[55] and missing data for certain residues, we subjected the allosteric correlation matrix to a Gaussian low-pass filter (Fig. 5b). The most dramatic correlations were found in residues belonging to the C-terminal helix α5 (Fig. 5b, e region V), indicating that most residue pairs in this helix respond in a correlated fashion to the seven mutations. This finding held true even when the A113P dataset, which is the data collected on the protein with a mutation in a residue in α5, was removed from the calculation. Thus, the C-terminal helix appears to be sensitively coupled to the rest of CheY.

We next utilized Ohm to predict the inter-residue correlations within CheY (Fig. 5c). As we used unphosphorylated protein for

the CHESCA, we also used the unphosphorylated state of CheY (PDB ID: 1JBE) for Ohm analysis. There were 12 regions in common in the matrices obtained from CHESCA and Ohm. Regions I, II, III, IV, and V each involve consecutive residues (Fig. 5e): region I consists of residues 6 to 9; region II consists of residues 33 to 43; region III consists of residues 50 to 58 and includes the allosteric site residue D57; region IV consists of residues 85 to 107, including T87 and Y106, which are the two critical residues in the dominant allosteric pathway (Fig. 5e, green cylinders); and region V consists of residues 112 to 124, which are all in the C-terminal helix, corroborating the speculation that the C-terminal helix is coupled to the rest of CheY. Besides, the coupling in the C-terminal helix can also be demonstrated by Anisotropic Network Model (ANM) through ANM 2.1 server[56]. The other seven regions involve correlations between residues

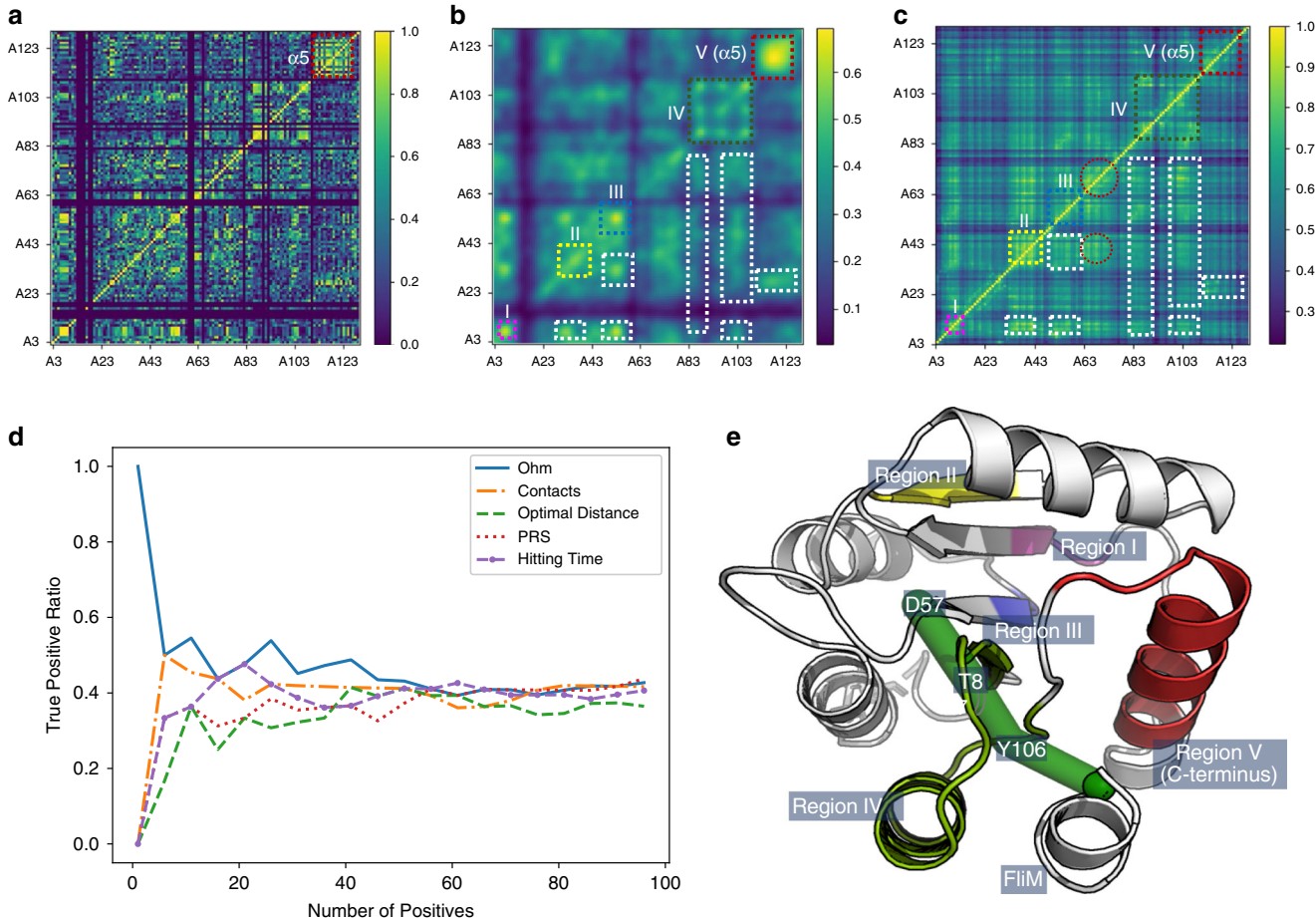

**Fig. 5 Comparison of allosteric correlations measured by CHESCA and predicted by Ohm. a** Allosteric correlations measured by CHESCA. Blank rows and columns are caused by missing data for residues 1, 2, 13, 14, 15, 16, 18, 19, 59, 60, 61, 65, 82, 90, 92, and 110. CHESCA data are in Supplementary Data 1. **b** Allosteric correlations measured by CHESCA subjected to Gaussian filtration. Dashed boxes, including the colored boxes (I, II, III, IV, and V) and the white boxes, are common regions to CHESCA and Ohm analysis. **c** Allosteric correlations predicted by Ohm. Regions in dashed circles only appear in allosteric correlations predicted by Ohm. **d** The true-positive ratio of allosteric correlations predicted by Ohm, native contacts, the shortest paths lengths, PRS, and the hitting time. Data of all methods for the comparison can be found in Supplementary Data 3. **e** Tertiary structure of CheY Regions indicated in panels **b** and **e** are labeled. The green cylinders represent the allosteric pathway.

located in regions I, II, III, IV, and V. The similarities of the matrices obtained from CHESCA and Ohm indicate the ability of Ohm to reliably identify inter-residue allosteric correlations in proteins.

Since PRS[39,40], the hitting time in Chennubhotla and Bahar's Markov propagation model[34,35], and the shortest paths lengths have also been used[38] to evaluate the correlation between residues in proteins, we further quantitatively compared Ohm to these methods, as well as the native contacts (Fig. 5d). We ranked all residue pairs by the predicted correlation and selected a certain number of top-ranked residue pairs. We then measured TPR—the ratio of the number of residue pairs that match CHESCA results (true positives) to the total number of selected residue pairs (positives)—to evaluate the accuracy of the calculated correlations. A selected residue pair matches CHESCA results when the corresponding CHESCA correlation is higher than 0.5. The TPR decreases as the number of positives increases, while the TPR of Ohm is higher than other methods when the number of positives is less than 60. When 20 top-ranked residue pairs are selected, the TPR of Ohm and the hitting time are both 47.6%, while the TPR of native contacts, shortest paths lengths, and PRS are 38.1%, 33.3%, and 33.3%, respectively.

**Stability of allosteric pathways over an MD simulation**. The perturbation propagation algorithm relies only on an input static structure, but protein structures are dynamic. Therefore we analyzed whether the conformational changes induced by in the CheY protein structure during the time course of a molecular dynamics (MD) simulation[57] caused significant variation in the allosteric network. We used Gromacs to perform a 100-ns molecular dynamics simulation of CheY using the crystal structure (PDB ID: 1F4V) as the starting structure. We extracted five snapshots at 0, 25, 50, 75, and 100 ns, and used Ohm to analyze the allosteric pathways. The results indicated that the allosteric network of CheY changes moderately over the simulation time course but that the core allosteric pathway, D57-T87-Y106-FliM, remains stable (Supplementary Fig. 25). Only at 75 ns, were T87 and Y106 not the most critical residues, but even at this time point the pathway D57-T87-Y106-FliM existed. In addition, W58, I95, G105, and K109 are among the critical residues in allosteric pathways in all five snapshots. Thus, although Ohm uses only the static tertiary structure of a protein as input, under conditions where conformational change is induced the allosteric pathway calculated by Ohm is stable. This observation is consistent with our proposed view of allostery, whereby protein

structure is coupled to dynamics and the allosteric phenomenon is a manifestation of this coupling.

**Factors that affect perturbation propagation.** As the perturbation propagation algorithm is de facto a random process, the results of each calculation differ. Hence, to evaluate the statistical significance of the perturbation propagation algorithm outcome, we performed multiple perturbation propagation simulations and ensure convergence of the results. To determine the number of steps required for convergence and for the statistical error to reach an acceptable level, we tested the standard deviation of ACI when different numbers of perturbation propagation rounds were used (Supplementary Fig. 26A). As the number of perturbation propagation rounds was increased from 100 to 10,000, the standard deviation of ACI decreased (Supplementary Fig. 26B). The standard deviations of ACI values were negligible when the number of perturbation propagation rounds was 10,000.

To interrogate the difference of using different experimental structures as input for Ohm analysis, we calculated ACI of residues in CheY by using four different experimental structures: 1FQW (*apo* CheY-BeF$_3^-$), 3CHY (*apo* unphosphorylated CheY), 1F4V (CheY-BeF$_3^-$-FliM), and 2B1J (unphosphorylated CheY-FliM). Because of the very short lifetime of phosphorylated CheY, its structure is not available. BeF$_3^-$ can bind at the allosteric site of CheY to mimic the phosphorylation effect to finally result in the binding of FliM at the active site. We observe that BeF$_3^-$ in 1FQW and 1F4V both have the highest ACI values (Supplementary Fig. 27). On the other hand, the ACI peak at the allosteric site D57 is much more prominent in *apo* structures than in *holo* structures. In the unphosphorylated *holo* structure (2B1J), the ACI peak at the allosteric site is even less prominent than other regions. Based on this result, we propose a four-state hypothesis for CheY. In the unphosphorylated state of CheY, multiple regions are allosterically correlated with the active site, including the allosteric site. When CheY is activated by phosphorylation or the binding with BeF$_3^-$, the allosteric site is so strongly affecting the active site that it finally leads to the binding with FliM. After the binding event, the allosteric correlation in the allosteric site becomes less prominent. When the *holo* structure is unphosphorylated, the allosteric correlation at allosteric site is even lower than other regions so as to protect the *holo* structure from being separated by any remotely propagated perturbation at the allosteric site.

Another factor that largely affects the performance of allosteric site prediction performance is the selection of residues in the active site to serve as the start of the propagation algorithm. We calculated ACI of residues in CheY-FliM complex (1F4V) by setting all residues in FliM as the start of the algorithm, and also calculated ACI of residues in *apo* CheY structure (3CHY) by setting all residues, residue Y106, or K119 on the CheY-FliM binding surface as the start of the propagation algorithm, respectively. By using FliM as the start for the bound structure (Supplementary Fig. 28A), or using all binding surface residues as the start for the unbound CheY structure (Supplementary Fig. 28B), we can successfully find the allosteric site D57. However, if we choose Y106 (Supplementary Fig. 28C) or K119 (Supplementary Fig. 28D) for unbound CheY structure, we cannot find the allosteric site D57. Thus, we recommend to use all residues on the active site for the unbound structure.

Parameter $\alpha$ in Eq. (3) is a user-defined parameter to amplify or reduce probability. We evaluated the dependence of the predictions as a function of $\alpha$ on the allosteric interaction network of CheY (Supplementary Fig. 29). When $\alpha$ was increased from 1 to 5, the absolute values of ACI increased. However, the positions of the peaks in ACI did not change. The network of

allosteric interactions within CheY changed only slightly, and the most important pathway, D57-T87-Y106-FliM, was unaffected, and the most critical residues remained T87 and Y106. Thus, the value of $\alpha$ does not affect the accuracy of the results. To further determine the value of $\alpha$ that can result in the best performance, we calculate ACI of residues in CheY by using $\alpha$ from 0.5 to 10 (Supplementary Fig. 30). When $\alpha$ is 10, the ACI values of nearly all residues are approaching 1, driving the identification of ACI hotspots intractable. Therefore, we prefer low $\alpha$-values (0.5–3) than high $\alpha$-values (>3).

For proteins, the backbone–backbone atom contacts between two sequence-adjacent residues are excluded when computing the perturbation propagation probability matrix. If contacts between backbone atoms of two adjacent residues are not excluded, perturbation propagates mainly through the backbone from one residue to its sequence-adjacent residues, which generates a large number of non-dominant pathways, and the identification of genuine allosteric pathways becomes prohibitively inefficient. When versions of Ohm excluding or including backbone–backbone contacts between sequence-adjacent residues were used to predict the allosteric pathways of CheY, only the version that excluded backbone–backbone contacts identified the correct allosteric site and allosteric pathways (Supplementary Fig. 31).

**Determining the impact of mutations on allosteric pathways.** Although allosteric pathways are insensitive to protein dynamics, mutations on certain residues, taking CheY as an example[45,58–62], especially those located in core allosteric pathways, can influence the allosteric behavior of proteins. To interrogate to what extent the mutations on peripheral residues, which are irrelevant to allosteric pathways, influence coupling between the allosteric site and the active site, we used Ohm to identify the allosteric pathways of two CheY mutants, V21A and I55V. The tertiary structures of these two mutants were generated using Eris[63]. The Ohm analysis showed that the allosteric pathways in these two mutants changed moderately relative to that in the wild-type protein (Supplementary Fig. 32). The core pathway, D57-T87-Y106-FliM, remained dominant in both mutants, however. The ACI values between the active site and the allosteric site are 0.2, 0.3, and 0.3 in the wild-type protein and V21A and I55V mutants, respectively. Thus, surprisingly, the allosteric correlations in the two mutants are even stronger than that in the wild-type structure.

To validate this result, we performed autophosphorylation experiments with wild-type CheY and the two mutants. CheY can either be phosphorylated by its sensor kinase CheA or autophosphorylated in the presence of an appropriate small-molecule phosphodonor[47]. Autophosphorylation results in a change in the fluorescence emission intensity at 346 nm, which derives from W58; this is a useful measure of functional activity[45,59]. When the data were analyzed, the two CheY mutants, V21A and I55V, both had higher values, by about 50%, of the slope of the plot of the observed rate constant versus phosphodonor concentration (Supplementary Fig. 33), which is consistent with the Ohm predictions.

## Discussion
By definition, allostery involves the propagation of signals between sites in a protein structure through a network of residues[22,64,65]. Hence Ohm, which is built upon a network modeling approach, is intuitively appealing. The benefit of network models is that they are solely dependent on protein structure, and thus they yield solutions more rapidly and cost-effectively than models that rely on molecular dynamics simulations. In network methods that are based on Markov model, the

transition matrix (or the conditional probability matrix) is normalized to satisfy the condition that the total sum of the probability of each residue with respect to other residues equals to 1, which is the premise of the theoretical derivation to directly calculate ACI or other metrics of residue correlations. However, in our propagation probability matrix, this condition is not satisfied, so we develop a propagation algorithm to calculate ACI. In actuality, satisfying this condition implies that a perturbation on one residue could only propagate to one of its contacted residues, which is not similar to the case of a real protein, where a residue can propagate perturbation to multiple contacted residues or even not propagate.

Since allosteric response has also been found in mechanical networks[66] and physical principles allowing for allosteric communication at distant sites have been studied in an artificial network[67], we hypothesize that allosteric phenomena are a general property of heterogeneous media and that coupling between distal sites of the heterogeneous material propagates through regions of higher coupling density. Packing heterogeneities in proteins result in specific propagation of perturbations between distal sites within the structure, thus resulting in allosteric coupling. The finding that allostery is an intrinsic property of heterogeneous protein structure opens avenues for engineering of allosteric protein switches and developing allosteric drugs and also sheds light on the relationship between protein structure and dynamics. Ohm derives perturbation propagation probability from residue–residue contacts instead of atom-level energies; this is probably the reason that the network identified by Ohm analysis was stable despite alterations in protein structure over the time course of a molecular dynamics simulation. In addition, since the packing density is higher around residues that have higher numbers of inter-atomic contacts, perturbation is likely to propagate through these higher density regions, and the number of contacts is a proxy for the probability of signal propagation through these residues.

Although here we used Ohm for analysis of interaction networks within proteins, the underlying perturbation propagation algorithm can be extended to other biomolecules, such as RNA, and even to heterogeneous materials. The calculation of the perturbation propagation probability matrix will bear great significance in extending Ohm to other materials. For proteins, the selection of an appropriate distance cutoff of contacts, the exclusion of backbone–backbone atom contacts between two sequence-adjacent residues, and the formula that converts contacts to probability all play crucial parts in computing the perturbation propagation probability matrix. To extend Ohm to identify RNA allosteric sites and pathways we will interrogate the relationship between allosteric communication and different types of inter-residue interactions, such as base stacking, to construct an appropriate strategy for calculation of the perturbation propagation probability matrix.

In summary, we have developed a model that can predict allosteric sites, pathways, and inter-residue correlations. We backward-validated the performance of Ohm by successfully mapping allosteric networks in a dataset composed of 20 allosterically regulated proteins for which high-resolution structures are available and the allosteric sites are known. We forward-validated the ability of Ohm to predict inter-residue allosteric correlations by comparing Ohm predictions with NMR CHESCA measurements for the protein CheY. We further ascertained the impact of dynamics and mutations on allosteric pathways. We anticipate that Ohm will be an essential tool for protein allostery analysis in drug discovery and protein engineering.

## Methods

**Average atom-contacts matrix**. From a 3D protein structure, the Ohm algorithm first extracts all the atom-wise contacts. Two atoms within 3.4 Å are counted as a

contact. The distances between every two atoms in the protein structure are then calculated:

$$C_{ij} = \sum_{a,b} H\left(r_0 - \left|\vec{r_a^i} - \vec{r_b^j}\right|\right), \tag{1}$$

where $C_{ij}$ is the number of atom contacts between residue $i$ and residue $j$, and $a$ and $b$ are atoms in residues $i$ and $j$, respectively. $a$ and $b$ cannot be backbone atoms simultaneously if $|i - j| = 1$. $r_0$ is the distance cutoff, $\vec{r_a^i}$ is the position of atom $a$ in residue $i$, and $\vec{r_b^j}$ is the position of atom $b$ in residue $j$. $H(x)$ is the Heaviside step function.

Subsequently, the number of contacts each atom in residue $i$ forms with atoms in residue $j$ is determined by dividing the number of contacts between residues $i$ and $j$ by the number of atoms in residue $i$. Likewise, we divide the number of contacts between residues $i$ and $j$ by the number of atoms in residue $j$ to evaluate how many contacts each atom in residue $j$ forms with atoms in residue $i$:

$$N_{ij} = \frac{C_{ij}}{C_i}, N_{ji} = \frac{C_{ij}}{C_j}, \tag{2}$$

where $N_{ij}$ is the number of average atom-contacts of residue $i$ with respect to residue $j$; $C_{ij}$ is the number of contacts between residue $i$ and residue $j$; $C_i$ is the number of atoms in residue $i$; and $C_j$ is the number of atoms in residue $j$. $C_{ij}$ is always equal to $C_{ji}$, whereas $N_{ij}$ is not necessarily equal to $N_{ji}$.

**Perturbation propagation probability matrix**. Based on the average atom-contacts matrix, the perturbation propagation probability matrix is calculated:

$$\begin{aligned} P_{ij} &= 1 - p_{ij} = 1 - e^{-\alpha \cdot N_{ij}} \\ P_{ji} &= 1 - p_{ji} = 1 - e^{-\alpha \cdot N_{ji}} \end{aligned}, \tag{3}$$

where $P_{ij}$ is the probability that the perturbation of residue $i$ will be propagated to residue $j$; $p_{ij}$ is the probability that the perturbation of residue $i$ will not be propagated to residue $j$; $\alpha$, currently set to 3.0, is a user-defined parameter to amplify or reduce probability.

**The perturbation propagation algorithm**. First, a vector (**V**) of size $N$ is built, where $N$ is the number of residues in the structure. If residue $i$ undergoes a conformational change, the $i$th element $V_i$ is assigned a value of 1, otherwise it is assigned a value of 0. Then, another vector (**W**) of size $N$ is built. Elements in **W** are all initially assigned values of 0. A third vector (**B**) is built to store the neighbors of all residues. Element $B_i$ is a set consisting of all the residues that have contacts with residue $i$. In the fourth vector (**T**) of size $N$, all elements are assigned values of 0.

Next, each of the residues in the active site is assigned 1. For instance, if the residue in the active site is residue $n$, both $V_n$ and $W_n$ are set to be 1. Subsequently, based on $B_n$, all neighbors of residue $n$ are identified. If $m$ is one of the neighbors of residue $n$, a random number $r$ is generated, and if $r < P_{nm}$ the value of $V_m$ is set to 1, otherwise the value of $V_m$ is set to 0. No matter what $V_m$ is, $W_m$ is set to 1. All the neighbors of $m$ as then identified and values in **V** and **W** are determined. This process is repeated until all values in **W** are 1. Then, $T_i$ is added by 1 if $V_i$ has been assigned a value of 1. For the next round, **V** and **W** are cleared, and the process is repeated $10^4$ times. Finally, vector **T** is normalized. $T_i$ is the value of allosteric coupling intensity of residue $i$ with respect to residue n. The perturbation propagation process is illustrated in Supplementary Fig. 34.

In order to identify allosteric sites, the residues in the active sites are excluded and the residue with the highest ACI among the remaining residues is selected as the allosteric site. The propagation algorithm is slightly adjusted to identify the allosteric pathways between the active site and the allosteric site. A stack **S** is constructed to store all the pathways that pass through the active site and the allosteric site. Suppose residue $n$ and residue $m$ are the active site and the allosteric site, respectively. Starting from residue $n$, the propagation process is performed, and the current path is added to **S** if the end of the path is $m$. The propagation is performed $10^4$ times and the histogram of all paths is statistically evaluated and stored in **S**. The path identified most often is the most likely allosteric pathway.

Each allosteric pathway is assigned a specific weight, which is a measure of its importance in allosteric communication. Based on these weights, we identify critical residues in allosteric pathways. Suppose $\{p_i\}$ is the collection of all pathways containing residue $a$, where $p_i$ represents the importance of allosteric pathway $i$. We use $p_a$ to represent the importance of residue $a$, and we set the initial value of $p_a$ to zero. Then, for each of the pathways in the collection $\{p_i\}$, we update the value of $p_a$ according to the equation below:

$$p_a = p_a + p_i - p_a * p_i. \tag{4}$$

When the importance of all pathways has been substituted into the equation, the value of $p_a$ is the final importance of residue $a$.

**Allosteric hotspots identification**. We first calculate the distances between all residue pairs in the protein and initialize a distance matrix $M(i, j)$, where $i$ and $j$ are residue indices. The distance between two residues is defined as the minimum distance between their atoms. For each residue, its neighbors $G(i)$ are then identified

by a distance threshold of 4.5 Å. A vector **D** (direction) is then initialized with a size of $N$-the number of residues. $D(i)$ is then assigned by the index of the neighbor residue that has a higher ACI than residue $i$. If there is no neighbor with a higher ACI, then $D(i)$ is assigned by $-1$. This way, each residue that has a direction value of $-1$ represents an allosteric hotspot. All other residues in each allosteric hotspot can be found by looking for residues that are finally directing to the hotspot center.

**Molecular dynamics simulation.** Gromacs[68] was employed to conduct molecular dynamics simulations of CheY (PDB ID: 1F4V). The Amber99SB-ILDN[69] force field was used in tandem with the TIP3P water model in a cubic periodic box. The structure was solvated using water molecules, and sodium counterions were added to neutralize residue charges. A short initial energy minimization of 50,000 steps was performed to resolve steric clashes. The temperature of 300 K was maintained using a V-rescale thermostat with a coupling constant of 0.1 ps. The solvent density was adjusted under isobaric and isothermal conditions at 1 bar and 300 K. A Parrinello-Rahman barostat with isotropic pressure coupling and a coupling constant of 0.1 ps was used to set the pressure at 1 bar.

**Protein expression and purification.** A plasmid containing *E. coli* CheY was provided by Dr. Robert Bourret (University of North Carolina at Chapel Hill) and was subcloned into the pET28a plasmid (Novagen). All mutants were prepared by site-directed mutagenesis. The CheY vector was transformed into BL21 Star (DE3) cells (Invitrogen) and grown on M9 minimal media with the appropriate isotope (s): $^{15}NH_4Cl$ (99%) and/or D-glucose (U-$^{13}C_6$—99%) as the sole nitrogen and carbon sources, respectively. Cells were grown at 37 °C and induced with 1 mM isopropyl 1-thio-β-D-galactopyranoside when the $OD_{600}$ reached 0.6 and were grown for an additional 22–26 h at 20 °C. The cells were harvested by centrifugation, resuspended in buffer A (25 mM Tris, 10 mM $MgCl_2$, pH 8.0), and sonicated. The lysate was then centrifuged at 6000 r.p.m. and dialyzed overnight into buffer A at 4 °C. The protein was purified on a Q-Sepharose Fast Flow column (GE Healthcare) equilibrated with buffer A and eluted in buffer B (buffer A with the addition of 1.5 M NaCl). Q-Sepharose purified protein was passed over a G75 superdex gel-filtration column equilibrated with NMR buffer (50 mM NaP$_i$, 0.02% $NaN_3$, pH 7.0 and an appropriate amount of $MgCl_2$ and/or EDTA). The CheY elution peak was concentrated for further experiments.

**CheY autophosphorylation assays.** Changes in tryptophan fluorescence were used to monitor CheY autophosphorylation kinetics. Protein was transferred to assay buffer (100 mM HEPES, 10 mM $MgCl_2$, pH 7.0) by passing purified CheY through a G-25 gel-filtration column equilibrated with assay buffer, prior to dilution of CheY to 5 μM. Phosphoramidate solutions were mixed with CheY to final concentrations of 5, 15, 25, 50, and 75 μM phosphoramidate phosphodonor, with a final CheY concentration of 2.5 μM. Enough KCl was added to reach an ionic strength of 200 mM for each reaction. Fluorescence measurements at 25 °C for F8V, V21A, I55V, and WT were made using a Luminescence Spectrometer LS50B[70]. Tryptophan fluorescence intensity was measured with an excitation wavelength of 292 nm, an emission wavelength of 346 nm, an emission slit width of 10 nm. Autophosphorylation assays were conducted by mixing CheY with each phosphoramidate solution and measuring fluorescence intensity over time until the intensity remained constant. Kinetic curves at each concentration of phosphoramidate were fit to three parameter exponential decay curves ($I = y_0 + ae^{-bt}$, where $I$ = fluorescence intensity, $b = k_{obs}$, and $y_0$ is I at $t = \infty$). Plots of $k_{obs}$ vs. [phosphoramidate] were fit to lines, with the slope equal to $k_{phos}/K_s$ and the $y$-intercept equal to $k_{dephos}$. Curve-fitting was carried out using SigmaPlot.

**NMR assignments.** All NMR spectra were collected on 1 mM CheY samples in NMR buffer (50 mM NaP$_i$, 0.02% $NaN_3$, pH 7.0,10 mM $MgCl_2$ and 10% $^2H_2O$). NMR spectra were recorded at 15 °C on Varian INOVA spectrometers equipped with room-temperature (500 and 600 MHz) or cryogenic (700 MHz) probes. Triple resonance experiments were carried out to assign wild-type CheY in the presence of 10 mM $Mg^{2+}$. Assignments for mutants were typically made by comparison to wild-type spectra, but in select cases HNCACB and CBCA(CO)NH datasets were recollected for the mutants.

**CHEmical Shift Covariance Analysis (CHESCA).** CHESCA analysis was performed on chemical shifts of unphosphorylated wild-type CheY and the seven mutants, F8V, D13K, M17A, V21H, T87I, Y106W, and A113P. Chemical shifts were observed for pseudo-phosphorylated (i.e., bound to phosphomimic BeF$_3^-$) wild-type and mutant forms of CheY, these were discarded from the analysis. As CheY undergoes a significant conformational change upon phosphorylation, inclusion of these forms resulted in CHESCA results that simply reflected that conformational change. By contrast, CHESCA on unphosphorylated (or inactive) forms should reveal only pre-existing inter-residue correlations present in the single dominant form of CheY. NMR assignments were made for nearly all residues in all CheY variants. In cases where an assignment could not be made for a residue of a particular mutant, rather than removing the residue from analysis, that assignment was approximated by replacement with the wild-type chemical shift. With very few missing assignments and a large number of variants analyzed, such minor errors were not expected to affect the results.

$^1$H and $^{15}$N chemical shifts were referenced, converted into combined chemical shifts (CCS), and arranged in a matrix to calculate CHESCA correlations. As the data were originally collected without CHESCA in mind, exact external $^1$H referencing was not carried out, and thus referencing was accomplished by internal re-referencing of a few variants to yield a self-consistent dataset. CCS values are calculated as

$$CCS(i) = H(i) + 0.15*N(i), \quad (5)$$

where $H(i)$ and $N(i)$ are the $^1$H and $^{15}$N chemical shifts of the $i^{th}$ residue, respectively. CHESCA was carried out as originally described[52,53] using in-house Matlab scripts. The correlation matrix of CCS values for all the residues was prepared using the matlab command "corr" and hierarchical cluster tree was calculated by matlab "linkage" command where clustering was done using complete linkage along with the "absolute correlation" metric option.

**Reporting summary.** Further information on research design is available in the Nature Research Reporting Summary linked to this article.

## Data availability
The data that support the findings of this study are in the Supplementary Data including both the experimental data and the computational data. The experimental data include the Fluorescence assay data (Supplementary Data 1) and the NMR CHESCA data (Supplementary Data 2); the computational data include the Ohm analyses of all 20 proteins (Supplementary Data 3), the comparison to Amor's method (Supplementary Data 4), and the correlation analysis of CheY (Supplementary Data 5). We have also provided the analyses results of the 20 proteins in the Ohm website (http://ohm.dokhlab. org). The links for these analyses are in Supplementary Table 3. All other data are available from the corresponding author on reasonable request.

## Code availability
Ohm is available at http://ohm.dokhlab.org. All source code is provided at https://bitbucket.org/dokhlab/ohm.

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

## Acknowledgements

We thank Dr. David Mowrey for his initial contribution to the project, Emily Frieben for her proofreading, Konstantin Popov for initial dataset preparation, Maria S. McGresham for preparation of NMR samples and assistance with NMR analysis, and Bradley T. Falk and Jeffrey P. Bonin for CHESCA MATLAB code. We acknowledge support from the National Institutes for Health (5R01GM123247, 2R01 GM114015, and 1R35 GM134864 to N.V.D.) and the Passan Foundation. The project described was also supported by the National Center for Advancing Translational Sciences, National Institutes of Health, through Grant UL1 TR002014. The content is solely the responsibility of the authors and does not necessarily represent the official views of the NIH.

## Author contributions

N.V.D. designed and conceived the project. N.V.D. and J.W. proposed the perturbation propagation algorithm. J.W. performed all the computational work, including the implementation of the algorithm, the design of the website, and all the computational test and analyses. A.L. and A.J. designed the experiments. L.R.M. created all the mutants and most of the NMR samples for CheY. C.G. made samples and analyzed data for V21A CheY and performed the autophosphorylation experiments. A.J. and A.L. analyzed the NMR and CHESCA data. J.W. and N.V.D. wrote most part of the manuscript. A.J. and A.L. wrote the experimental part of the manuscript.

## Competing interests

The authors declare no competing interests.
