## [Peer Review File · Nature Communications]

Reviewers' comments:

Reviewer #1 (Remarks to the Author):

The authors propose a structure-based stochastic computational method to map allosteric sites, pathways and networks based on the hypothesis that packing heterogeneities serve as a mechanism to relay allosteric perturbations. The proposed software was tested using a dataset of 20+ allosteric structures and was experimentally validated through mutations, enzymatic assays and NMR CHESCA of CheY. Ohm offers a platform to interpret and design allosteric regulatory switches. Being non-MD based, Ohm is considerably faster than alternative computational methods that rely on lengthy MD trajectories and given its speed, it is likely that Ohm will find many valuable applications in bioinformatics.

Main Suggestions:

- The authors suggest that Ohm is more reliable than prior computational methods, such as that by Amor (reference 22). A direct Ohm vs. Amor's method comparison would be helpful. Some elements of such comparison are present in the ms, but a dedicated section would be useful to highlight experimental data explained by one method but not the other.

-Please explain why the performance of Ohm decreases when active sites are buried or allosteric sites are close to active sites.

-Please provide clearer guidelines on how to find optimal ranges of the alpha parameter in equation 3

-Specify which structure should be used by Ohm as input. Typically allostery involves transitions between at least two structures (e.g. unbound or bound to allosteric effectors or inhibitors, phosphorylated or in general with or without PTMs), so it'd be helpful to know which structure is required by Ohm. This point is in part touched for CheY. However, offering general guidelines to Ohm users in this regard will enhance the impact of this software.

-The authors show that the core elements of the allosteric network predicted by Ohm for CheY are largely preserved along a 100 ns MD trajectory. Since multiple structures are available for CheY, how do the Ohm results change for different experimental CheY input structures?

-Explain more clearly how the Ohm perturbations used for CheY mimic the perturbations used in CHESCA NMR

Minor suggestions:

-The abstract can be shortened, especially in the initial part on the relevance of allostery

-Define more specifically 'heterogeneous media' in abstract. In fact, this aspect is not fully developed in the ms and is more part of the speculative section of the discussion. It'd be better if the abstract more closely reflected the actual results of the ms.

-Fig3 PDB code needs to be changed to 1F4V.

-Color codes need to be explained in Fig S27

Reviewer #2 (Remarks to the Author):

General Comments

Wang et al. present a computational methodology to uncover allosteric pathways in proteins. The method proposed, Ohm, uses the contact topology of a protein 3D structure to create propagation matrices P_{ij} that when evaluated using stochastic perturbations provides insights on allosteric relevance of residues. The method relies on the knowledge of active sites to be used as starting points in the network perturbation process. The authors analyzed 20 protein systems using Ohm and

correctly identified residues involved in allosteric communication for 13 of them. For the remaining 7, the Allosteric Coupling Indices were significant but not necessarily the top ones. The authors also used the proposed technique to create residue-residue correlation networks that were compared against chemical shift covariance analysis in the CheY response regulator. The authors propose that this methodology does not rely in expensive molecular dynamics simulations and might serve as a useful and user friendly tool to study allostery, compared to other methods.

In general, I think the methodology and the presentation are relevant and the implementation seems easy to use, e.g. the Ohm website is user friendly. However, in order to be convinced of scientific advantages of this methodology, I would like to see more quantitative analysis and controls as well as clear comparison with other methods, for example the Amor et al. paper referred several times in the study. This would help the authors make the point that there are other advantages than computational complexity or user friendliness. I would like to see a revision that could answer the following questions:

1. Is the dataset of 20 proteins diverse enough to support the generality claims of the study? If so, the authors should describe this in the article as opposed to only mention that this set was compiled in another study.
2. Could the authors also include some negative controls to show that proteins that are not known to be allosteric would not have enriched ACI values? One premise of the article is that any protein has the potential for allostery, but we know that the functional relevance of allostery is different depending on the system. It would be good to see results showing these negative controls.
3. When looking at the definition of the propagation matrix, it was not clear to me why is it important to normalize the number of contacts per atom. In other words, why would a single atom with X contacts will give a higher probability than two atoms that form the same amount of contacts? To me these two cases are more or less equivalent since ultimately the number of interactions is the same. I understand that the idea is that the density of contacts is important, but I would expect that in terms of residues but not necessarily in terms of the number of atoms per side chain. Could the authors clarify this?

4. In some cases, allostery is realized not by single monomers but via oligomerization, have the authors tried these types of cases? Would they expect that the methodology works in the same way? If so, could the authors include one of such cases? This would make the application of the methodology much more general.

5. For the Caspase-1 results, the authors mentioned perfect accord with experimental results. I think it is important to quantify the agreement, maybe with correlation coefficient or other metrics, as opposed to a less objective qualitative metric.

6. For the CheY system, the authors mention that D57 is an allosteric site. My understanding is that the phosphorylation site should be considered an “active site” as opposed to an allosteric site. Since this is a residue that is modified and then the effect of this modification is propagated to the allosteric site. How much would the results change if D57 is used as an active site instead? Would the method also identify the allosteric site?

7. This work references the work of Amor et al. in several instances, both for selecting the systems to work on as well as to compare specific results. However, it is not clear to me from reading the paper, if Ohm is better at predicting allosteric sites and pathways than the methodology proposed by Amor et al. which was also validated in other 80 proteins. The authors should give more details in this respect because both seem to be graph theoretical approaches, dealing with the same systems but there is no general comparison in this work.

8. When comparing inter-residue allosteric correlations from Ohm and CHESCA, it was not really clear to me why the need of a low-pass filter. The article references some experimental errors, but then, why did the authors also applied the filter to the computational results? Similar to my previous comment, it is important to have a more quantitative comparison as opposed to a qualitative assessment of the agreement. Particularly if this is supposed to be used as an experimental validation.

9. For the comparisons with CHESCA, I am wondering how would this comparison would look like if we were to use only the native contacts in the protein. Would these dominate? The authors could use this comparison as a test case that their method reflects more than just structural topology, and if not, then they should revise their conclusions. Finally, is CHESCA considered a ground truth for allosteric correlations? Since the results are fairly affected by conformational transitions, I am not sure the authors should state that “...the predicted allostery in CheY protein predicted by Ohm was validated by NMR CHESCA studies”. I would re-state that the inter-residue correlations are correlated with CHESCA analysis, but I am not sure about being validated.

10. The authors described how the method reaches convergence after 10^4 iterations. The propagation matrix seems similar to a transition matrix, although I am not sure if it has all the properties. However, if the properties are met, maybe there is no need to run stochastic simulations, but just check for stationary vectors. I might be making some wrong assumptions, but it would be good if the authors check this possible connection with Markovian properties of a transition matrix.

11. The discussion states that Ohm is a reverse process of direct coupling analysis. I am not sure if they are really comparable. On the one side, Ohm uses contacts as input to then create a dynamical view of allostery, while DCA identifies direct interactions that are important during evolution. These definitions are not necessarily reverse. In fact, one could build a “DCA matrix” and maybe run it through the Ohm algorithm and possibly find something related to allostery. Rather than reverse, I would suggest that DCA could be used as input instead of the contact map, although I am not sure what the results would look like.

12. My understanding is that the algorithm starts in the active site and then it ends at the allosteric site. However, for the case of the last repressor, DNA is defined as the “active site”. If you see Figure S12, the caption says that the active site is the red circle, i.e. close to DNA, however the pathway in Fig. S12D starts near residue A79 which is the black square or allosteric site. It feels to me that there might be some confusion here that must be clarified.

13. The agreement with CHESCA depends on the conformational state of the protein. It is important to mention this limitation in the discussion section. There might be cases where the conformational state is not as clearly defined as in the case of phosphorylation.

Specific comments

1. Table S1 includes active and allosteric site codes that are not clearly defined, please include the definition and positions of these residues.

2. The authors mention that residues with high SASA have larger ACI values and then they propose this due to a protective environment for the allosteric pathway. But to me, this is just a consequence of the definition of P_{ij} , where the probability for propagation is lower for residues with less contacts as is the case of higher SASA residues. I am not completely convinced by the protection hypothesis, could the authors elaborate on this or maybe remove the claim?

3. Figure S1 seems to have an error in the caption. The first sentence seems like it should not be there.

4. Page 12, "atom-average contacts" sounds a bit awkward, I would rather use "average atom-contacts"

5. Line 295, replace "residue l" with "residue i"

6. Line 314, replace "by1" with "by 1"

Reviewer #3 (Remarks to the Author):

The manuscript by Wang et al. introduces a network-based approach to identify, (i) allosteric sites in a protein, (ii) allosteric pathways which locate critical residues, and (iii) allosteric correlations between residues. The network is constructed based-on the number of contacts between residue pairs, but the value is normalized by the number of atoms in a residue so that the attributes of the links are not symmetrical. Using these values to calculate probability of information transfer between contacting nodes, selected perturbations are propagated along the network for a large number of cycles whereby convergence is achieved and the most visited sites/paths are determined.

At the outset, I wish to point out that the web-based interface that displays the results of these analyses is well-thought out and is very user friendly if the results are biologically relevant. This is where I question the methodology and the significance of the findings.

Network-based models to study path and community analyses in relation to allostery have long been in use in the protein structure community. Of course, there are different ways to construct the network on which the communication pathway between binding site and allosteric site is to be studied; yet, the main features of the network construction approach outlined in this work has to be

put into perspective in relation to the existing methodologies. In particular, early pioneering work by Vishveshwara et al. has been overlooked in this manuscript.

Once the network construction approach has been into perspective for the potential user, the authors should also rigorously compare their findings to existing methods.

For example, Chennubhotla and Bahar used the normalized adjacency matrix element as the probability that a random walker uses to jump between connected nodes and to define communities (please see Mol. Syst. Biol. 2006). In fact, once the transition probabilities are in place, one can in principal calculate the converging values of the random walk, without actually doing the 10000 rounds of simulations (in fact, convergence is displayed in figure S22). Why would the approach here be superior to that methodology? I note that, there is some comparison to Amor's method for the CheY example, but the origin of the differences observed is not discussed. Finally, the guarantee of convergence issue also makes me question the "perturbation propagation" label of the methodology as all the pathways are pre-determined by the probabilities on the links.

On the other hand, in relation to finding pathways connecting allosteric to the active site, the results outlined in the manuscript resemble the findings from a simple optimal path analysis where the network is constructed via proximity of the residues, and the link weights are selected from contact potentials (please see Atilgan, Turgut, Atilgan, Biophys. J. 2007). One would definitely like to see a comparison of the findings here, with those from such a simplistic analysis (e.g., on page 5, R286 and E390 are found as the most critical residues, but they also reside on the shortest path connecting the two end points). Note that, that approach also automatically takes into account the problem of including $(i,i+1)$ contacts mentioned on page 9 and figure S24.

One also wonders if the critical residues residing along the pathways found in this work have high betweenness centrality, another graph theoretical parameter that has been found to be useful in identifying hot spots in proteins. I was able to locate the mention of this quantity to the prediction of hot spots in proteins in a paper that appeared as early as in 2005 (del Sol et al, Bioinformatics) as well as many others that appeared later on. The authors acknowledge in the Discussion (page 10) that the perturbation likely propagates through higher density regions, which is conceptually similar.

In terms of response of folded protein structures to perturbations, there is at least one network-based perturbation-response analysis method that is widely used to determine effector and sensor residues for external perturbations as well as important connections for signaling within a protein that is not mentioned in the manuscript (please see <http://prody.csb.pitt.edu/prs/>).

I think all these approaches will provide similar results, for the basic reason that the folded protein structure and residue cross-correlations dominate the phenomena investigated in this line of research. Case-in-point: The CHESCA results displayed in figure 5A,B are mainly controlled by residue interactions, manifested in the adjacency matrix (which, I presume will resemble the perturbation propagation probability matrix derived here). This is why, the results in Figure 5D,E are similar to, e.g. Anisotropic Network Model (ANM) findings (I have used the online tool ANM 2.1 to roughly compare the highlighted regions). Again, the authors need to persuade the readers why a more complex analysis is necessary to find results whose main features are captured by the neighbor information. Since the central example protein here, CheY, is relatively small, it may not be a good system to demonstrate the subtleties of the current approach; its surface to volume ratio is high and there are only a few densely packed residues which can reside on allosteric pathways.

Minor points:

Page 2: Sentence beginning, "It follows that if ..." needs rewording.

Page 5: The sentence, "These low-ACI areas serve to protect high-ACI areas from the influences of perturbation from the surrounding environments," is speculative.

Page 9: On the impact of mutations on the results, how different are the structures of the mutants from the wild type? Do the results vary significantly more than those obtained at different time points in the MD simulation and presented in figure S21?

Page 10: The term "thermodynamically linked" is vague.

Page 13: Subsection explaining allosteric hotspots identification is extremely hard to follow.

Figure 2C and all similar figures; please also specify the ID of the start/end points.

Figure 3, caption: Please correct the PDB ID.

Figure 5, caption: Please reword "columns in are caused..."

Figure S23: Please use the alpha symbol as in the main manuscript. Also, what appears in the three columns of this figure need to be explained in the caption.

Figure S27: This figure also needs explanation. What is displayed in not self-evident.

Reviewer #1:

The authors propose a structure-based stochastic computational method to map allosteric sites, pathways and networks based on the hypothesis that packing heterogeneities serve as a mechanism to relay allosteric perturbations. The proposed software was tested using a dataset of 20+ allosteric structures and was experimentally validated through mutations, enzymatic assays and NMR CHESCA of CheY. Ohm offers a platform to interpret and design allosteric regulatory switches. Being non-MD based, Ohm is considerably faster than alternative computational methods that rely on lengthy MD trajectories and given its speed, it is likely that Ohm will find many valuable applications in bioinformatics.

Main Suggestions:

- The authors suggest that Ohm is more reliable than prior computational methods, such as that by Amor (reference 22). A direct Ohm vs. Amor's method comparison would be helpful. Some elements of such comparison are present in the ms, but a dedicated section would be useful to highlight experimental data explained by one method but not the other.

Reply: *Thank you very much for your valuable suggestions. We have now added the comparison between Ohm and Amor's method (Results section, Supplementary Table 4, and Supplementary Figure 22).*

-Please explain why the performance of Ohm decreases when active sites are buried or allosteric sites are close to active sites.

Reply: *Initially, Ohm clustered residues on protein surface based on ACI values because we assumed the allosteric site to be on the surface. Thus, if the allosteric sites are buried, they would be ruled out from the final candidates. On the other hand, if the allosteric sites are close to the active sites, residues at these two locations can be clustered together readily. Therefore, we have updated the residue clustering algorithm by discarding the limit of residues on protein surface, so the performance of Ohm will not decrease when active sites are buried. We have also excluded active sites from ACI calculation so the active site and allosteric site would not be clustered together when they are in close proximity. We have now completely re-evaluated the performance of Ohm on the 20 protein data set (Results section).*

-Please provide clearer guidelines on how to find optimal ranges of the alpha parameter in equation 3

Reply: *We have now elaborated more on how to find the optimal range of alpha parameter in the results section. In general, we prefer low alpha values than high alpha values.*

-Specify which structure should be used by Ohm as input. Typically allostery involves transitions between at least two structures (e.g. unbound or bound to allosteric effectors or inhibitors, phosphorylated or in general with or without PTMs), so it'd be helpful to know which structure is required by Ohm. This point is in part touched for CheY. However, offering general guidelines to Ohm users in this regard will enhance the impact of this software.

Reply: *We have now added detailed analysis of the choice of input structure in the results section. We generally recommend using the unbound structure as the input and select all residues on the binding surface as the start of the propagation algorithm.*

-The authors show that the core elements of the allosteric network predicted by Ohm for CheY are largely preserved along a 100 ns MD trajectory. Since multiple structures are available for CheY, how do the Ohm results change for different experimental CheY input structures?

Reply: *We have now added the analysis of Ohm predictions when different experimental structures are used as input. We performed allosteric analysis for CheY by using 4 different experimental structures: 1FQW (apo CheY-BeF₃⁻), 3CHY (apo unphosphorylated CheY), 1F4V (CheY-BeF₃-FliM), and 2B1J (unphosphorylated CheY-FliM). Because of the very short lifetime of phosphorylated CheY, its structure has been inaccessible. BeF₃⁻ can bind at the allosteric site of CheY to mimic the phosphorylation effect to finally result in the binding of FliM at the active site. In Supplementary Figure 27, we can see that BeF₃⁻ in 1FQW and 1F4V both have the highest ACI values. On the other hand, the ACI peak at the allosteric site D57 is much more prominent in apo structures than in holo structures. In the unphosphorylated holo structure, the ACI peak at the allosteric site is even less prominent than other regions. Based on this result, we propose a four-state hypothesis for CheY. In the unphosphorylated state of CheY, multiple regions are allosterically correlated with the active site, including the allosteric site. When CheY is activated by phosphorylation or the binding with BeF₃⁻, the allosteric site is so strongly affecting the active site that it finally leads to the binding with FliM. After that, the allosteric correlation in the allosteric site becomes less prominent and finally when the holo structure is unphosphorylated, the allosteric correlation at allosteric site is even lower than other regions to protect the holo structure from being separated by any remotely propagated perturbation at the allosteric site.*

-Explain more clearly how the Ohm perturbations used for CheY mimic the perturbations used in CHESCA NMR

Reply: *We have now added a quantitatively analysis of Ohm perturbations in the result section and Figure 5 by calculating the true positive ratio.*

Minor suggestions:

-The abstract can be shortened, especially in the initial part on the relevance of allostery

Reply: *We have now shortened the abstract.*

-Define more specifically 'heterogeneous media' in abstract. In fact, this aspect is not fully developed in the ms and is more part of the speculative section of the discussion. It'd be better if the abstract more closely reflected the actual results of the ms.

Reply: *We have now moved the 'heterogeneous media' sentences to the results section. Regarding heterogeneous media, we posit that the allosteric phenomenon is a physical effect of heterogenous media. Condensed matter physics and statistical mechanics are physical subjects that study heterogenous media. We refer to the concepts developed in those fields, as the methods developed in this work are inspired by the established physical models (Rocks, JW, et al. PNAS.2017.114: 2520-2525; Yan, L, et al. PNAS. 2017. 114:2526-2531).*

-Fig3 PDB code needs to change to 1F4V.

Reply: *Corrected.*

-Color codes need to be explained in Fig S27

Reply: *We have now added the explanation in the caption of Fig S34 (the old Supplementary Figure 27).*

Reviewer #2:

General Comments

Wang et al. present a computational methodology to uncover allosteric pathways in proteins. The method proposed, Ohm, uses the contact topology of a protein 3D structure to create propagation matrices P_{ij} that when evaluated using stochastic perturbations provides insights on allosteric relevance of residues. The method relies on the knowledge of active sites to be used as starting points in the network perturbation process. The authors analyzed 20 protein systems using Ohm and correctly identified residues involved in allosteric communication for 13 of them. For the remaining 7, the Allosteric Coupling Indices were significant but not necessarily the top ones. The authors also used the proposed technique to create residue-residue correlation networks that were compared against chemical shift covariance analysis in the CheY response regulator. The authors propose that this methodology does not rely in expensive molecular dynamics simulations and might serve as a useful and user-friendly tool to study allostery, compared to other methods.

In general, I think the methodology and the presentation are relevant and the implementation seems easy to use, e.g. the Ohm website is user friendly. However, in order to be convinced of scientific advantages of this methodology, I would like to see more quantitative analysis and controls as well as clear comparison with other methods, for example the Amor et al. paper referred several times in the study. This would help the authors make the point that there are other advantages than computational complexity or user friendliness. I would like to see a revision that could answer the following questions:

1. Is the dataset of 20 proteins diverse enough to support the generality claims of the study? If so, the authors should describe this in the article as opposed to only mention that this set was compiled in another study.

Reply: *Thank you very much for your valuable suggestions! The dataset was compiled by Amor and coworkers from SCOP database, which is a manually curated database using a hierarchical classification scheme collecting protein domains into structurally similar groups. The major classes of cytoplasmic proteins in the database are α , β , α/β , $\alpha+\beta$, and multi-domain, covering all the major fold-types for cytosolic proteins. They randomly selected proteins from each of the five classes. The dataset includes monomer, dimer, trimer, tetramer, hexamer, and even dodecamer proteins. The ligands of these proteins include chemical compounds, nucleosides, peptides, and DNA molecules. The lengths of proteins range from 147 to 3311 amino acids. Thus, the dataset covers a broad region of protein structure space and we think it's diverse enough for our test. We have now added the description of this dataset in the manuscript.*

2. Could the authors also include some negative controls to show that proteins that are not known to be allosteric would not have enriched ACI values? One premise of the article is that any protein has the potential for allostery, but we know that the functional relevance of allostery is different depending on the system. It would be good to see results showing these negative controls.

Reply: *We have now added a designed four-helix bundle protein (IMFT) as the negative control (Supplementary Figure 24). The tertiary structure colored by the calculated ACI values shows that apart from the designated pseudo “active site” – the N terminal – there is no ACI hotspots in the structure.*

3. When looking at the definition of the propagation matrix, it was not clear to me why is it important to normalize the number of contacts per atom. In other words, why would a single atom with X contacts will give a higher probability than two atoms that form the same amount of contacts? To me these two cases are more or less equivalent since ultimately the number of interactions is the same. I understand that the idea is that the density of contacts is important, but I would expect that in terms of residues but not necessarily in terms of the number of atoms per side chain. Could the authors clarify this?

Reply: *Our network is in the residue level, so the reason that we normalize the number of contacts is because we assume smaller residues are more easily affected than larger residues when they have the same amount of contacts. Suppose when we run a residue-level coarse-grained MD simulation, if the net forces applied to two residues are equal, then the residue that has a smaller mass would have a larger acceleration, which makes it more likely to be affected.*

4. In some cases, allostery is realized not by single monomers but via oligomerization, have the authors tried these types of cases? Would they expect that the methodology works in the same way? If so, could the authors include one of such cases? This would make the application of the methodology much more general.

Reply: *There are 7 monomers, 2 dimers, 1 trimer, 7 tetramers, 2 hexamers, and 1 dodecamer. We have now added this information to Supplementary Table 1 and also in the manuscript.*

5. For the Caspase-1 results, the authors mentioned perfect accord with experimental results. I think it is important to quantify the agreement, maybe with correlation coefficient or other metrics, as opposed to a less objective qualitative metric.

Reply: *We have now added a quantitative analysis by calculating the true positive ratio (TPR) of Ohm predictions with respect to CHESCA results in the results section and Figure 5D. We have also uses this quantitative analysis to compare Ohm predicted correlations to native contacts, shortest paths lengths, and two other methods.*

6. For the CheY system, the authors mention that D57 is an allosteric site. My understanding is that the phosphorylation site should be considered an “active site” as opposed to an allosteric site. Since this is a residue that is modified and then the effect of this modification is propagated to the allosteric site. How much would the results change if D57 is used as an active site instead? Would the method also identify the allosteric site?

Reply: *We think that the perturbation in protein is passing from the allosteric site to the active site (Dokholyan, NV. Chem Rev. 2016. 116:6463-6487), so the phosphorylation site was considered as the allosteric site. We have now performed the allosteric analysis for CheY by using D57 as the active site. The results show that the allosteric site, which binds the FliM in this case, is corresponding to an ACI peak, which is identified as an allosteric hotspot (Supplementary Figure 1A). Residue 87 and 106 are still identified as the two most critical residues in the allosteric pathways (Supplementary Figure 1B).*

7. This work references the work of Amor et al. in several instances, both for selecting the systems to work on as well as to compare specific results. However, it is not clear to me from reading the paper, if Ohm is better at predicting allosteric sites and pathways than the methodology proposed by Amor et al. which was also validated in other 80 proteins. The authors should give more details in this respect because both seem to be graph theoretical approaches, dealing with the same systems but there is no general comparison in this work.

Reply: *We have now added a detailed comparison of Ohm with Amor's method in Supplementary Table 4, Supplementary Figure 22, and the results section.*

8. When comparing inter-residue allosteric correlations from Ohm and CHESCA, it was not really clear to me why the need of a low-pass filter. The article references some experimental errors, but then, why did the authors also applied the filter to the computational results? Similar to my previous comment, it is important to have a more quantitative comparison as opposed to a qualitative assessment of the agreement. Particularly if this is supposed to be used as an experimental validation.

Reply: *Thank you for pointing this out. We have now removed the use of low-pass filter for Ohm analysis. We also added a quantitative analysis of Ohm predictions by calculating the true positive ratio (TPR) with respect to CHESCA measurements in the results section and Figure 5D.*

9. For the comparisons with CHESCA, I am wondering how would this comparison would look like if we were to use only the native contacts in the protein. Would these dominate? The authors could use this comparison as a test case that their method reflects more than just structural topology, and if not, then they should revise their conclusions. Finally, is CHESCA considered a ground truth for allosteric correlations? Since the results are fairly affected by conformational transitions, I am not sure the authors should state that "...the predicted allostery in CheY protein predicted by Ohm was validated by NMR CHESCA studies". I would re-state that the inter-residue correlations are correlated with CHESCA analysis, but I am not sure about being validated.

Reply: *We have now added the comparison of Ohm predictions to native contacts, the shortest paths lengths calculated by Dijkstra's algorithm, and two other methods. We calculated the true positive ratio (TPR) with respect to CHESCA measurements. The TPR of Ohm is much higher than that of the other methods.*

CHESCA is not considered as a ground truth for allosteric correlations because it has several drawbacks. First, it is derived from covariance analysis of chemical shifts of only several mutants, while the accuracy could be improved when data from more mutants are included in the analysis. Second, there are missing data for several residues. Third, there exist experimental errors. All these factors make CHESCA not an ideal ground truth, so we only selected residue pairs that have CHESCA correlation > 0.5 as 'true' correlated residue pairs when we calculated TPR.

Finally, we have re-stated the sentence with "...the predicted allostery in CheY protein predicted by Ohm was found correlated with NMR CHESCA studies" as suggested.

10. The authors described how the method reaches convergence after 10^4 iterations. The propagation matrix seems similar to a transition matrix, although I am not sure if it has all the properties. However, if the properties are met, maybe there is no need to run stochastic simulations, but just check for stationary vectors. I might be making some wrong assumptions,

but it would be good if the authors check this possible connection with Markovian properties of a transition matrix.

Reply: *A transition matrix is usually normalized so that the total of transition probability from a state i to all other states must be 1, but the propagation matrix is different from the transition matrix in that the total of the probability of a residue affecting other residues is not necessarily to be 1. This is why we cannot derive ACI theoretically from the initial perturbation probability matrix.*

11. The discussion states that Ohm is a reverse process of direct coupling analysis. I am not sure if they are really comparable. On the one side, Ohm uses contacts as input to then create a dynamical view of allostery, while DCA identifies direct interactions that are important during evolution. These definitions are not necessarily reverse. In fact, one could build a “DCA matrix” and maybe run it through the Ohm algorithm and possibly find something related to allostery. Rather than reverse, I would suggest that DCA could be used as input instead of the contact map, although I am not sure what the results would look like.

Reply: *Thank you very much for pointing this out. Since the DCA is not very relevant with our manuscript, we finally decided to delete this discussion.*

12. My understanding is that the algorithm starts in the active site and then it ends at the allosteric site. However, for the case of the last repressor, DNA is defined as the “active site“. If you see Supplementary Figure 12, the caption says that the active site is the red circle, i.e. close to DNA, however the pathway in Fig. S12D starts near residue A79 which is the black square or allosteric site. It feels to me that there might be some confusion here that must be clarified.

Reply: *The allosteric site prediction algorithm starts in the active site and ends in other nodes in the network because users usually know the active site and want to know the position of the allosteric site. But the pathways identification algorithm starts in the allosteric site and ends in the active site because the perturbation in proteins is propagating from the allosteric site to the active site. We have now clarified it in the manuscript.*

13. The agreement with CHESCA depends on the conformational state of the protein. It is important to mention this limitation in the discussion section. There might be cases where the conformational state is not as clearly defined as in the case of phosphorylation.

Reply: *Thank you so much for pointing this out. We have now performed comprehensive analysis of ACI in different conformation states of CheY. We performed Ohm analysis for CheY by using 4 different experimental structures: 1FQW (apo CheY-BeF₃⁻), 3CHY (apo unphosphorylated CheY), 1F4V (CheY-BeF₃-FliM), and 2B1J (unphosphorylated CheY-FliM). Because of the very short lifetime of phosphorylated CheY, its structure has been inaccessible. BeF₃⁻ can bind at the allosteric site of CheY to mimic the phosphorylation effect to finally result in the binding of FliM at the active site. In Supplementary Figure 27, we can see that BeF₃⁻ in 1FQW and 1F4V both have the highest ACI values. On the other hand, the ACI peak at the allosteric site D57 is much more prominent in apo structures than in holo structures. In the unphosphorylated holo structure, the ACI peak at the allosteric site is even less prominent than other regions. Based on this result, we propose a four-state hypothesis for CheY. In the unphosphorylated state of CheY, multiple regions are allosterically correlated with the active site, including the allosteric site. When CheY is activated by phosphorylation or the binding with BeF₃⁻, the allosteric site is so strongly affecting the active site that it finally leads to the binding with FliM. After that, the allosteric*

correlation in the allosteric site becomes less prominent and finally when the holo structure is unphosphorylated, the allosteric correlation at allosteric site is even lower than other regions to protect the holo structure from being separated by any remotely propagated perturbation at the allosteric site.

Specific comments

1. Supplementary Table 1 includes active and allosteric site codes that are not clearly defined, please include the definition and positions of these residues.

Reply: *We have now added the details of residues in active and allosteric sites in Supplementary Table 1.*

2. The authors mention that residues with high SASA have larger ACI values and then they propose this due to a protective environment for the allosteric pathway. But to me, this is just a consequence of the definition of Pij, where the probability for propagation is lower for residues with less contacts as is the case of higher SASA residues. I am not completely convinced by the protection hypothesis, could the authors elaborate on this or maybe remove the claim?

Reply: *Thank you for the suggestion and we have now removed the claim.*

3. Supplementary Figure 1 seems to have an error in the caption. The first sentence seems like it should not be there.

Reply: *We have now removed the first sentence.*

4. Page 12, “atom-average contacts” sounds a bit awkward, I would rather use “average atom-contacts”

Reply: *Thank you very much for this suggestion. We have replaced all occurrences of “atom-average contacts” with “average atom-contacts”.*

5. Line 295, replace “residue I” with “residue i”

Reply: *Corrected.*

6. Line 314, replace “by1” with “by 1”

Reply: *Corrected.*

Reviewer #3:

The manuscript by Wang et al. introduces a network-based approach to identify, (i) allosteric sites in a protein, (ii) allosteric pathways which locate critical residues, and (iii) allosteric correlations between residues. The network is constructed based-on the number of contacts between residue pairs, but the value is normalized by the number of atoms in a residue so that the attributes of the links are not symmetrical. Using these values to calculate probability of information transfer between contacting nodes, selected perturbations are propagated along the network for a large number of cycles whereby convergence is achieved and the most visited sites/paths are determined.

At the outset, I wish to point out that the web-based interface that displays the results of these analyses is well-thought out and is very user friendly if the results are biologically relevant. This is where I question the methodology and the significance of the findings.

Network-based models to study path and community analyses in relation to allostery have long been in use in the protein structure community. Of course, there are different ways to construct the network on which the communication pathway between binding site and allosteric site is to be studied; yet, the main features of the network construction approach outlined in this work has to be put into perspective in relation to the existing methodologies. In particular, early pioneering work by Vishveshwara et al. has been overlooked in this manuscript.

Reply: *Thank you very much for your valuable suggestions! We have now introduced other network models, including Vishveshwara's method, in the introduction section and also outlined the main features of our network model in the discussion section.*

Once the network construction approach has been into perspective for the potential user, the authors should also rigorously compare their findings to existing methods.

Reply: *We have now added a detailed comparison between our method and Amor's method (Supplementary Table 4, Supplementary Figure 22). We also compared Ohm to native contacts, shortest paths lengths, PRS, and Chennubhotla and Bahar's Markov propagation approach.*

For example, Chennubhotla and Bahar used the normalized adjacency matrix element as the probability that a random walker uses to jump between connected nodes and to define communities (please see Mol. Syst. Biol. 2006). In fact, once the transition probabilities are in place, one can in principal calculate the converging values of the random walk, without actually doing the 10000 rounds of simulations (in fact, convergence is displayed in Supplementary Figure 22). Why would the approach here be superior to that methodology? I note that, there is some comparison to Amor's method for the CheY example, but the origin of the differences observed is not discussed. Finally, the guarantee of convergence issue also makes me question the "perturbation propagation" label of the methodology as all the pathways are pre-determined by the probabilities on the links.

Reply: *Chennubhotla and Bahar proposed a Markov propagation-based approach for elucidating the potential pathways of allosteric communication. In a Markov model of residue network, the transition matrix is normalized to satisfy the condition that the sum of the probability of each residue with respect to other residues equals to 1. However, in our propagation probability matrix, this condition is not satisfied and this is the reason we need to develop a new algorithm instead of theoretical derivation. Satisfying this condition implies that a perturbation from one residue could only be propagated to one another residue at a time, while in our model a residue can propagate perturbation to multiple residues at a time. Therefore, this is the difference between Ohm and the Markov propagation method. We have now added the explanation in the discussion section.*

On the other hand, in relation to finding pathways connecting allosteric to the active site, the results outlined in the manuscript resemble the findings from a simple optimal path analysis where the network is constructed via proximity of the residues, and the link weights are selected from contact potentials (please see Atilgan, Turgut, Atilgan, Biophys. J. 2007). One would definitely like to see a comparison of the findings here, with those from such a simplistic analysis (e.g., on page 5, R286 and E390 are found as the most critical residues, but they also reside on the

shortest path connecting the two end points). Note that, that approach also automatically takes into account the problem of including (i,i+1) contacts mentioned on page 9 and Supplementary Figure 24.

Reply: *Thank you very much for pointing this out. We tried to use Atilgan's program but it only gives the optimal path lengths rather than the actual pathways. They mentioned in their paper that they employed Dijkstra's algorithm to find the optimal pathways, so we implemented Dijkstra's algorithm to calculate the optimal pathways for all 20 protein and ranked the critical residues based on their betweenness centralities in the optimal pathways. The comparison results are in Supplementary Table 5.*

One also wonders if the critical residues residing along the pathways found in this work have high betweenness centrality, another graph theoretical parameter that has been found to be useful in identifying hot spots in proteins. I was able to locate the mention of this quantity to the prediction of hot spots in proteins in a paper that appeared as early as in 2005 (del Sol et al, Bioinformatics) as well as many others that appeared later on. The authors acknowledge in the Discussion (page 10) that the perturbation likely propagates through higher density regions, which is conceptually similar.

Reply: *Betweenness centrality is very similar to the importance of critical residues we proposed in the manuscript. We calculated the importance by iteratively employing Eq. 4, $p(a)=p(a)+p(i)-p(a)*p(i)$, where $p(a)$ is the importance of residue a and $p(i)$ is the probability of the i th pathway that passes through residue a . By definition, betweenness centrality can be calculated by $p'(a)=p'(a)+p'(i)$, iteratively. We have now calculated the betweenness centrality and ranked residues based on it. We compared it (column 3 in Supplementary Table 5) to the ranking results by the importance of residue (column 2 in Supplementary Table 5). For CheY (1F4V), residue 87 and 106 are 3rd and 4th ranked by betweenness centrality, while they are 1st and 2nd ranked by the importance of residue.*

In terms of response of folded protein structures to perturbations, there is at least one network-based perturbation-response analysis method that is widely used to determine effector and sensor residues for external perturbations as well as important connections for signaling within a protein that is not mentioned in the manuscript (please see <http://prody.csb.pitt.edu/prs/>).

Reply: *We have now added the introduction of PRS in the introduction.*

I think all these approaches will provide similar results, for the basic reason that the folded protein structure and residue cross-correlations dominate the phenomena investigated in this line of research. Case-in-point: The CHESCA results displayed in figure 5A,B are mainly controlled by residue interactions, manifested in the adjacency matrix (which, I presume will resemble the perturbation propagation probability matrix derived here). This is why, the results in Figure 5D,E are similar to, e.g. Anisotropic Network Model (ANM) findings (I have used the online tool ANM 2.1 to roughly compare the highlighted regions). Again, the authors need to persuade the readers why a more complex analysis is necessary to find results whose main features are captured by the neighbor information. Since the central example protein here, CheY, is relatively small, it may not be a good system to demonstrate the subtleties of the current approach; its surface to volume ratio is high and there are only a few densely packed residues which can reside on allosteric pathways.

Reply: *We have now calculated the true positive ratio (TPR) to compare Ohm predictions with PRS, Chennubhotla and Bahar's Markov propagation method, native contacts, and the shortest paths lengths derived from Dijkstra's algorithm (Figure 5D). The true positive ratio of Ohm is higher than other methods. I also mentioned in the manuscript that the C-terminal helix coupling could also be identified by ANM server.*

Minor points:

Page 2: Sentence beginning, "It follows that if ..." needs rewording.

Reply: *Reworded.*

Page 5: The sentence, "These low-ACI areas serve to protect high-ACI areas from the influences of perturbation from the surrounding environments," is speculative.

Reply: *We have now deleted this sentence.*

Page 9: On the impact of mutations on the results, how different are the structures of the mutants from the wild type? Do the results vary significantly more than those obtained at different time points in the MD simulation and presented in Supplementary Figure 21?

Reply: *We have now added the RMSD of mutants and different time frames of the MD simulation in the captions of Supplementary Figure 25 and S32, respectively.*

Page 10: The term "thermodynamically linked" is vague.

Reply: *Corrected.*

Page 13: Subsection explaining allosteric hotspots identification is extremely hard to follow.

Reply: *According to Reviewer 1's comments, we have updated the allosteric hotspots identification algorithm, which is essentially a residue clustering algorithm. We have completely rewrote the explanation of the allosteric hotspots identification algorithm in the methods section.*

Figure 2C and all similar figures; please also specify the ID of the start/end points.

Reply: *The start and end points are usually the ligands that bind to the active site and the allosteric site, respectively. The name of the ligands in these 20 proteins are in Supplementary Table 1. We did not put the name of the ligands in the start and end points because sometimes different kinds of ligands can bind to the active site or allosteric site.*

Figure 3, caption: Please correct the PDB ID.

Reply: *Corrected.*

Figure 5, caption: Please reword "columns in are caused..."

Reply: *Reworded.*

Supplementary Figure 23: Please use the alpha symbol as in the main manuscript. Also, what appears in the three columns of this figure need to be explained in the caption.

Reply: *Corrected.*

Supplementary Figure 27: This figure also needs explanation. What is displayed in not self-evident.

Reply: *Thank you the suggestion! We have now added the explanation.*

REVIEWERS' COMMENTS:

Reviewer #1 (Remarks to the Author):

The authors have adequately addressed our previous comments. Just a couple of very minor suggestions to enhance clarity:

p. 7 of merged PDF, lines 165-166: "These results indicate that the allosteric correlation between the allosteric site and the active site is generally reversible." We suggest rephrasing this statement to take into account that not all allosteric couplings are reversible as previously shown by E. Freire et al.

Fig. 5A-C: it would be very helpful to add a label for alpha5 in panels A-C of Fig. 5

Reviewer #2 (Remarks to the Author):

The new version of the manuscript, SI and the response to reviewers seem appropriate and have successfully answered my questions. I am particularly pleased with the effort to include more quantitative comparisons with other methods to demonstrate the advantage of this new methodology. I am also satisfied with the response to the other reviewers asking to clarify methodological aspects, acknowledge and properly cite previous work and the definition and clarification of the work in other systems.

Revisions were extensive and comprehensive with the possible drawback that the SI is possibly too long. I think this work deserves to be published and its original contributions are substantial.

Reviewer #3 (Remarks to the Author):

I have found the revised version of the manuscript by Wang et al. to be considerably improved. In particular, the relevant literature has now been more comprehensively covered and some of the computational details have been provided to enable the reproducibility of the findings. Moreover, the discussions are now based more on quantitative arguments and comparisons to additional measures that determine allosteric sites and residues that control allosteric pathways have been included.

Some typos:

line 352: allosteric response -> allosteric response

line 353: please reword, "...distant allosteric communication at distant site..."

Table S2: k_{cat} should not be capitalized

REVIEWERS' COMMENTS:

Reviewer #1

The authors have adequately addressed our previous comments. Just a couple of very minor suggestions to enhance clarity:

p. 7 of merged PDF, lines 165-166: "These results indicate that the allosteric correlation between the allosteric site and the active site is generally reversible." We suggest rephrasing this statement to take into account that not all allosteric couplings are reversible as previously shown by E. Freire et al.

Reply: *Thank you very much for your valuable suggestions. We have now rephrased this statement.*

Fig. 5A-C: it would be very helpful to add a label for alpha5 in panels A-C of Fig. 5

Reply: *We have now added the label in Figure 5.*

Reviewer #2

The new version of the manuscript, SI and the response to reviewers seem appropriate and have successfully answered my questions. I am particularly pleased with the effort to include more quantitative comparisons with other methods to demonstrate the advantage of this new methodology. I am also satisfied with the response to the other reviewers asking to clarify methodological aspects, acknowledge and properly cite previous work and the definition and clarification of the work in other systems.

Revisions were extensive and comprehensive with the possible drawback that the SI is possibly too long. I think this work deserves to be published and its original contributions are substantial.

Reply: *Thank you very much for your comments!*

Reviewer #3

I have found the revised version of the manuscript by Wang et al. to be considerably improved. In particular, the relevant literature has now been more comprehensively covered and some of the computational details have been provided to enable the reproducibility of the findings. Moreover, the discussions are now based more on quantitative arguments and comparisons to additional measures that determine allosteric sites and residues that control allosteric pathways have been included.

Some typos:

line 352: allostery response -> allosteric response

Reply: *Thank you very much for your comments! We have now corrected this typo.*

line 353: please reword, "...distant allosteric communication at distant site..."

Reply: *Reworded.*

Table S2: k_{cat} should not be capitalized

Reply: *Corrected.*